# Stimulus type shapes the topology of cellular functional networks in mouse visual cortex

Disheng Tang [1,2,3] ✉, Joel Zylberberg [4,5,8] ✉, Xiaoxuan Jia[1,3,6,8] ✉ & Hannah Choi [2,7,8] ✉

On the timescale of sensory processing, neuronal networks have relatively fixed anatomical connectivity, while functional interactions between neurons can vary depending on the ongoing activity of the neurons within the network. We thus hypothesized that different types of stimuli could lead those networks to display stimulus-dependent functional connectivity patterns. To test this hypothesis, we analyzed single-cell resolution electrophysiological data from the Allen Institute, with simultaneous recordings of stimulus-evoked activity from neurons across 6 different regions of mouse visual cortex. Comparing the functional connectivity patterns during different stimulus types, we made several nontrivial observations: (1) while the frequencies of different functional motifs were preserved across stimuli, the identities of the neurons within those motifs changed; (2) the degree to which functional modules are contained within a single brain region increases with stimulus complexity. Altogether, our work reveals unexpected stimulus-dependence to the way groups of neurons interact to process incoming sensory information.

Visual information is processed by networks of neurons spanning multiple regions of the neocortex. The interactions between these neurons determine the sensory information extracted by the brain and used to guide behavior. For this reason, much prior work has investigated properties of the networks that define the interactions between neurons in visual cortex. For example, some work has focused on the patterns of anatomical connectivity between individual neurons[1–4], or between larger voxels of cortical tissue[5–8]. At the same time, functional networks—which describe the interactions between neurons—can differ substantially from anatomical networks[9–11]. Notably, while anatomical connectivity is relatively fixed on the timescale of sensory processing, functional connectivity can vary as the neurons within the network adjust their firing patterns quickly to different stimuli[12,13]. This motivated us to ask whether and how different stimuli might engage different functional networks with single-neuron resolution within the

visual cortex. Despite the clear importance of this question for understanding visual processing, and the substantial literature on functional and anatomical neural network structures (reviewed below), we are unaware of any prior work that addressed how the topological structure of functional networks between individual neurons spanning multiple regions varies as the stimulus type changes. To fill this knowledge gap, we applied network analyses to simultaneous recordings from hundreds of neurons in mouse visual cortex. Our results indicate that distinct stimulus types can lead to different topological structures of functional networks between individual neurons in visual cortex.

Previous work investigated anatomical connectivity between cortical neurons and regions using electron microscopy[4,14], paired intracellular electrophysiology recordings[1,2], viral tracing[5,15], and diffusion tensor imaging[16]. These studies revealed many interesting

[1]School of Life Sciences, Tsinghua University, Beijing 100084, PR China. [2]Quantitative Biosciences Program, Georgia Institute of Technology, Atlanta 30332 GA, USA. [3]IDG/McGovern Institute for Brain Research, Tsinghua University, Beijing 100084, PR China. [4]Department of Physics and Astronomy, and Centre for Vision Research, York University, Toronto ON M3J 1P3 ON, Canada. [5]Learning in Machines and Brains Program, CIFAR, Toronto ON M5G 1M1 ON, Canada. [6]Tsinghua-Peking Center for Life Sciences, Tsinghua University, Beijing 100084, PR China. [7]School of Mathematics, Georgia Institute of Technology, Atlanta 30332 GA, USA. [8]These authors contributed equally: Joel Zylberberg, Xiaoxuan Jia, Hannah Choi. ✉e-mail: dishengtang3@gmail.com; joelzy@yorku.ca; jxiaoxuan@gmail.com; hannahch@gatech.edu

features of anatomical neuronal connectivity networks, like their modular organization and small-worldness[5,15,17,18], and their hierarchical structure[15]. While anatomical connectivity (e.g., synaptic connections between neurons) remains relatively static over the timescale of processing visual inputs, functional connectivity can be much more dynamic, thus motivating efforts to understand the relation between functional and anatomical connectivity[7,11,19–21]. These efforts are complicated by the fact that different types of stimuli lead to different dynamical patterns of neural activity and to different degrees of correlation between neurons[22–29]. Because functional connectivity depends on these properties–e.g., on the time-lagged correlation between the activities of neuron pairs[30,31]–the functional connectivity can depend on the stimulus presented in the experiment.

Despite this potential complication, stimulus- and task-related functional connectivity patterns obtained at a coarse scale using non-invasive functional magnetic resonance imaging (fMRI) have been reported to resemble resting-state functional connectivity patterns[32–34], while resting-state connectivity in turn resembles anatomical connectivity patterns[35]. In other reports–again, derived from fMRI experiments–stimulus-evoked functional interactions were found to vary with tasks or cognitive states[36–40]. These fMRI studies raised the important question of whether and how the functional connectivity of the underlying neuronal networks (i.e., at a finer single-neuron scale) might change with stimulus or task conditions.

Studying functional connectivity at this finer scale presents significant challenges due to technical limitations in simultaneously recording from large populations of neurons with high spatial and temporal resolution. Despite these challenges, prior work has shown that functional connectivity: (1) shows frequency dependency[41,42]; (2) varies by cell type within the cortex[43]; (3) depends on the contrast of a visual stimulus[44]; and (4) reflects the existence of two main groups of neurons, one whose activities follow those of the rest of the population, and one whose activities do not[26]. Other studies looked into assembly neurons[45], network dynamics[46], small-worldness[47,48] and rich-club structure[49]. While these studies have revealed much about the stimulus-dependence of functional networks at single-neuron resolution, they have not included detailed analyses of networks spanning multiple brain regions evoked by distinct stimulus types. On the other hand, the previous reports of network analysis applied to single-neuron resolution functional networks, either focused on responses mainly to drifting grating stimulus with spontaneous activity as a baseline comparison[30,31], thus precluding an assessment of stimulus-dependent network structure, or investigated the short-term adaptation of pairwise functional connections[50], lacking a comprehensive analysis of the whole network. Therefore, it is still unclear whether and how the topological organization of these functional networks (either within a brain region, or spanning multiple regions) depends on stimulus properties or other context-defining variables[51].

To fill this gap, we used network analysis methods (similar to those of refs. 30,31) to analyze the functional networks measured in response to 6 different types of stimuli, of varying degrees of complexity, ranging from full-field flashes up to natural movies. These networks were obtained from the simultaneously recorded activities of hundreds of neurons in 6 different cortical regions with implanted Neuropixels probes[30]. Thus, we were able to identify functional networks for each stimulus type, which spanned multiple brain regions. Note that to focus on between-stimulus analyses, we constructed one network based on all conditions for each stimulus type, hence the functional networks embody total correlations rather than signal or noise correlations. By studying the structures of these networks and how they varied with stimulus type, we identified several surprising features of the functional networks. First, while the distribution of different types of 3-neuron connectivity motifs were quite similar for the different stimuli, the specific identities of the neurons within those motifs depended on the stimulus. This means that the cortical network

is dynamically reorganized as the stimulus type changes, but does so in a manner that preserves the motif frequencies. This finding points to a potentially fundamental role for these motif distributions in maintaining the function of the cortical networks[52,53]. Secondly, we identified highly-interacting modules[54,55] and found that these modules were much more localized to a single brain region (as opposed to being distributed between regions) for stimuli with higher complexity, such as natural movies. Our results thus reveal distinct stimulus-dependent topology of cortical functional networks, and imply a key organizational principle underlying that stimulus-dependence: preserved relative motif frequencies.

## Results

To determine whether and how visual cortical functional networks depend on the stimulus presented to the animal, we analyzed data from Neuropixels probes inserted into six visual regions of mouse cortex (Fig. 1A: V1, LM, RL, AL, PM, AM), which is previously released by Allen Institute[30]. These probes simultaneously recorded neural activity from each of these six regions while the mice were presented with visual stimuli of varying degrees of complexity (Fig. 1B): flashes, drifting gratings, static gratings, natural scenes and movies, and gray screen (approximation for resting state, or spontaneous activity). From the responses to each stimulus, we extracted the directed functional connectivity using cross-correlograms (CCGs) between the spiking responses of pairs of neurons (Fig. 1C). In order to take polysynaptic connections into consideration[56], we examined 'sharp intervals' instead of the 'sharp peaks' that might be used to identify functions of monosynaptic connections[30,57,58]. These sharp intervals were defined to have a short latency and potentially multiple time lags, and were detected by searching for statistically significantly outlying values in the CCG. Identification of bidirectional connections was made possible by limiting lag $\tau$ to be non-negative, and each significant connection was defined as positive or negative depending on the sign of the significantly outlying CCG value (see Fig. 1C and Methods), similar to the definition used in previous work[59]. Intuitively, if the spiking of the source neuron is statistically strongly correlated with the firing or non-firing of the target neuron with a short time lag, then there exists a positive or negative functional connection between them. It is noteworthy that within our analyses, excitatory and inhibitory anatomical connections should be observed as positive and negative functional connections, respectively. However, it is essential to acknowledge that the reverse inference may not hold true. This is because, unlike effective or causal connections, functional connections reflect the co-occurrence of spiking activities rather than direct influence or causation.

To obtain a comprehensive understanding of the stimulus-dependent structure of the functional networks (Fig. 1D), we conducted network analyses at multiple topological scales, ranging from the properties of pairwise connections to the local connectivity patterns of third-order functional motifs, up to larger-scale functional modules. Our control analysis on running speed (Supplementary Fig. 1) showed that our subsequent observations are indeed determined by the stimulus and not by locomotion.

### Stimulus dependency of functional networks

We first investigated overall patterns of functional connections between neurons across stimulus types by comparing the functional connectivity matrices. We found there are some common network features observed across stimulus types. Specifically, the functional networks observed during all visual stimuli exhibited heavy-tailed degree distributions (Supplementary Fig. 2B). Networks with this property are known to be robust to random failures[60], however, they are more vulnerable to targeted attacks on hub neurons which could lead to reduced network efficiency as observed in Alzheimer's patients[61,62].

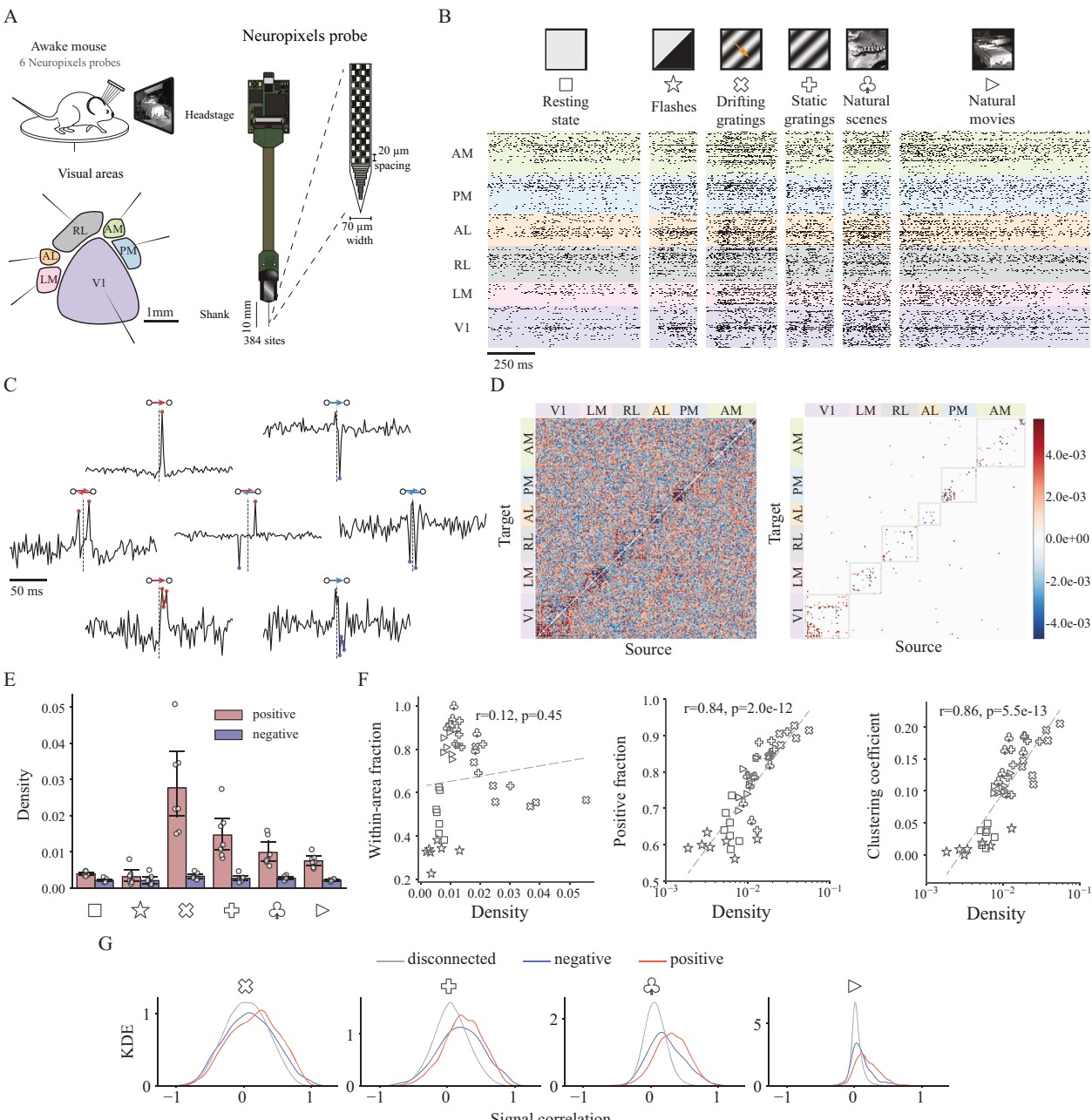

**Fig. 1 | From spike trains to functional connectivity for mouse visual cortex.**
**A** Schematic of data collection with Neuropixels probes inserted through six visual cortical areas (AM, PM, AL, RL, LM and V1), adapted from Jia, X. et al. High-density extracellular probes reveal dendritic backpropagation and facilitate neuron classification. Journal of neurophysiology 121, 1831-1847 (2019)[124]. **B** Example spike trains of 741 units from the visual cortex of a mouse during six different types of stimuli. For brevity, each stimulus type is denoted using a unique symbol in all figures. Image credit: Allen Institute for Brain Science. [https://observatory.brain-map.org/visualcoding/]. **C** Example jitter-corrected CCGs (cross-correlograms) of positive (red)/negative (blue), unidirectional/bidirectional and monosynaptic ("sharp peak")/polysynaptic ("sharp intervals") connections. **D** (left) Example matrix of jitter-corrected CCG with units ordered by area during natural movie stimuli. (right) Connectivity matrix with only significant connections ($|Z| > 4$). **E** Density of positive and negative connections during all visual stimuli. Density is defined as the number of connections normalized by total possible number of connections. Error bars represent 95% confidence interval, $n = 7$ mice. **F** Fraction of within-area connections, fraction of positive connections and clustering coefficient against network density. Each visual stimulus is characterized by a symbol, consistent with (**B**). $P$ values are obtained through two-sided Wald test. **G** Kernel density estimation (KDE) of signal correlation distributions for disconnected neuron pairs and pairs with negative/positive connections during presentations of four types of visual stimuli.

While functional networks show some shared characteristics like heavy-tailed degree distributions across stimuli, we also observed network properties vary with stimulus complexity. We found that natural stimuli (natural scenes and movies) tended to evoke fewer functional connections than grating stimuli (both static and drifting gratings) while full-field flashes drive the least correlated neural activities, on the same level as resting state activity (Fig. 1E). These findings are consistent with previous reports that natural stimuli decorrelate neurons in primary visual cortex (V1)[63,64]. While these previous works focus on V1, our results suggest that decorrelation by natural stimuli is a general property of cortical circuits: it is found in higher visual cortical areas as well.

While number of nodes (neurons) is predetermined by experimental recording, network density, defined as the present number of

connections normalized by the maximum possible number of connections, alone displays the most fundamental properties of a functional network. The differences in network density mainly originate from differences in positive connections (Fig. 1E), which results in the strong correlation between the fraction of positive connections and the network density (Fig. 1F, middle). We did a thorough analysis of the firing-rate dependence of functional connections and found that the difference in the number of connections during various stimulus types cannot be explained by the difference in firing rate (Supplementary Fig. 3). Even though natural stimuli do not evoke the densest functional networks, the fraction of within-area connections is largest for static gratings and natural stimuli (Fig. 1F, left, and Supplementary Fig. 2A; $p = 1.0 \times 10^{-3}$, Kolmogorov–Smirnov test, one-sided). This is closely related to the stimulus-dependent differences in modular network structure, which we analyzed in more details later in this paper.

To determine how the stimulus-dependence of the network density affects the network's topological structure, we measured the tendency for triplets of neurons to form closed triangles (e.g., three-neuron motifs 6,7,9-13 in Fig. 2C). This tendency is quantified by the clustering coefficient, and we found that it increases with increasing network density regardless of stimulus type (Fig. 1F, right).

Motivated by previous work showing that neurons with similar preferences tend to connect with each other[2,4,65], we compared the tuning similarity of neuronal pairs connected with positive and negative connections. To perform this comparison, we computed the kernel density estimation (KDE) for signal correlation during presentation of four visual stimulus types. Signal correlation is defined as the correlation between average responses of neurons to different stimulus conditions which is used to test whether two neurons have similar tuning curves[23,66]. We computed these signal correlation separately for pairs with positive connections, negative connections, and those with no connections (Fig. 1G). For natural movies, we regarded each frame as a different stimulus condition when computing the signal correlation[2]. Since there are only two conditions for flashes (dark or light), the signal correlation of either 1 or −1 could be trivial and thus is not considered in this analysis. Similarly, the signal correlation is ill-defined for the blank gray screen stimulus, and thus it was also omitted from this analysis.

For all visual stimuli, the signal correlations for connected neuron pairs tended to be larger than for disconnected pairs, which had distributions centered around zero (Fig. 1G; $p < 7.0 \times 10^{-150}$, rank-sum test, one-sided, adjusted using Benjamini/Hochberg method). Additionally, neurons with positive connections tended to have higher signal correlations than did pairs with negative connections (Fig. 1G; $p < 1.6 \times 10^{-3}$, rank-sum test, one-sided, adjusted using Benjamini/ Hochberg method).

In agreement with the previous findings that neurons close in space or sharing similar tuning curves are more likely to have synaptic connections[2,65], we found the probability of functional connections decreases with increasing distance and increases with their increasing signal correlation (Supplementary Fig. 4A, B; Cochran–Armitage test, two-sided). In addition, the probability of a functional connection being positive/negative significantly increased/decreased with signal correlation during all visual stimuli (Supplementary Fig. 4C, D; Cochran–Armitage test, two-sided), indicating that even though neurons with similar preferences generally tend to be connected, the sign of the connection depends on the extent of their tuning similarity.

Collectively, these analyses show that network density, fraction of connections that are within a brain region (as opposed to between regions), clustering coefficient, and the distribution of signal correlation, depend systematically on the stimulus type.

## Stimulus dependency of functional motifs

Having observed stimulus-dependency of the general network properties, we next turned our attention to the properties of the functional motifs. Specifically, we investigated two- and three-node motifs in the functional network. Similar to anatomically-defined structural motifs which form fundamental building blocks of neural circuits[1,67,68], functional motifs, which are defined by correlated neural activities, represent elementary information processing components of a functional network[69]. Although extensive over-representation of certain network motifs such as feedforward loop is found mostly in anatomical networks[1,52,67,70], functional motifs (especially triplets) also have been discovered to reflect the level of consciousness[71,72], accurately predict neural activity[73], encode global topological information[74], and exhibit coupling with anatomical connections[75].

To understand the distribution of two- and three-neuron motifs in each functional network, we adopted the intensity method for functional motif detection. This method computes the Z-score of the intensity for a given motif by comparing the motif frequency in the empirical network and in a randomly-generated surrogate network[76]. This method thus identifies how much more (or less) prevalent the motif is in the real network than would be expected by chance in a reference network randomly shuffled with certain preserved properties (e.g., density, degree distribution, etc) (Fig. 2A).

Note that, to characterize pairwise signed connections (i.e., signed two-neuron motifs), it is necessary to preserve the edge sign distribution when generating density-matched randomized surrogate networks[77]. For this reason, we included edge signs in the Pair-preserving model[67] that preserves the distribution of (n-1)-neuron motifs and used the resultant Signed-pair-preserving model to generate the surrogate network with the preserved signed (n-1)-neuron motifs for comparing the motif frequency between the true network and the randomly-generated one (Fig. 2A). Furthermore, we have consistently observed and validated our findings by maintaining the discretized distribution of anatomical distances between neurons (as shown in Supplementary Fig. 5). This rigorous approach underscores the robustness of our primary conclusions, affirming that they are an accurate representation of the inherent characteristics of the functional networks within the mouse visual cortex. In light of the limited data on the anatomical location of neurons, we have adopted the Signed-pair-preserving model for all analyses in the subsequent sections of our study.

As for two-neuron motifs, we found that bidirectional signed functional connections are much more frequent than would be expected by chance in the random network (Fig. 2B). This recapitulates the structural observations that bidirectional synaptic connections are highly over-represented in cortex relative to density-matched random networks[1].

We further studied distributions of three-neuron motifs in functional networks. There are 13 types of connected three-neuron motifs (Fig. 2C), of which the Feedforward Loop (FFL) is arguably the most studied type due to its ubiquitous nature in empirical networks such as gene systems and neuronal networks[70]. The lollipop plot in Fig. 2D shows the intensity Z-score obtained using the above intensity method for all types of signed three-neuron motifs ordered by their corresponding unsigned motif types. The colors in the plot represent each unsigned type (see Supplementary Fig. 6 for the full set of signed connectivity patterns). Interestingly, the most salient motifs were the same for the different visual stimuli. Specifically, the top six over-represented motifs are the same for different stimuli tested here (Fig. 2D, motif ID = p6, p9, p10, p11, p12, p13). In addition to these significantly over-represented motifs, the same three types of motifs were significantly under-represented for all stimuli (ID = p4, p5, p8). Interestingly, all 6 over-represented motifs contain at least one positive FFL (pFFL) structure (ID = p6, Fig. 2D) with the only difference among them being the number of mutual connections that accompany the pFFL structure. The last five of these over-represented motifs were studied without edge signs in previous works as 'mixed-feedforward-feedback loops' and have been found to be correlated to memory, as

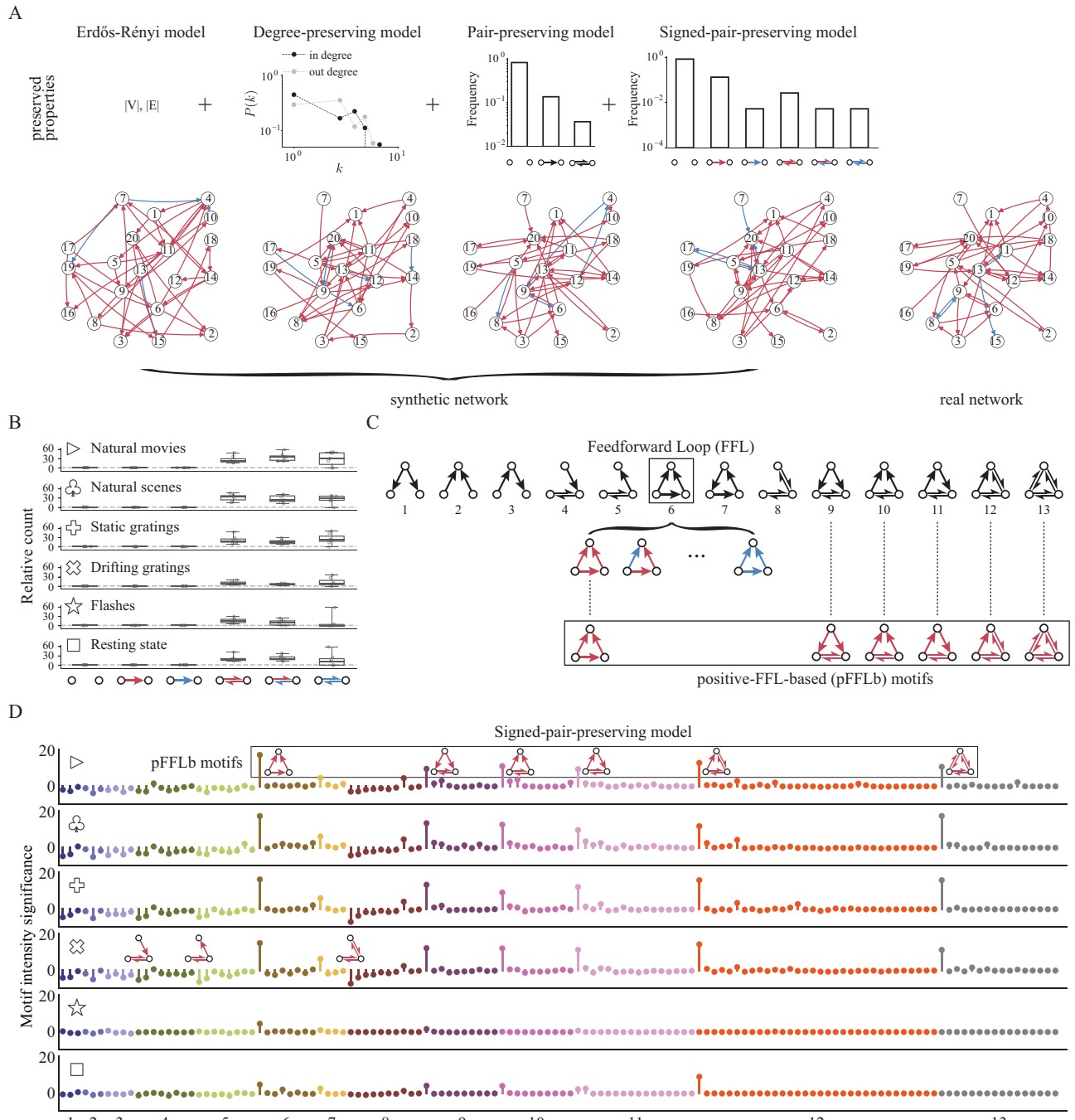

**Fig. 2 | Highly preserved local structure during different types of visual stimuli.**
**A** Illustration of the surrogate models: Erdős−Rényi model, Degree-preserving model, Pair-preserving model and Signed-pair-preserving model that exhibit a progressive increase in the number of preserved network properties. The preserved properties are the number of nodes (V)/edges (E), degree distribution, neuron-pair distribution and signed-pair distribution, respectively. The real network used to generate the synthetic networks is sampled from an example session during drifting gratings. **B** Relative count of signed neuron pairs using Erdős−Rényi model. All three types of bidirectional pairs are considerably over-represented. Box plots indicate median (middle line), 25th, 75th percentile (box) and minimum and maximum (whiskers), $n = 7$ mice. **C** 13 types of motifs without edge signs. Examples of signed motifs based on Feedforward Loop structure (motif ID = 6) are shown. Positive-FFL-based (pFFLb) motifs are defined as motifs with at least one FFL

structure consisting of all positive connections. Note that pFFLb motifs can be classified into four types based on the number of mutual connections: zero (ID = 6), one (ID = 9, 10, 11), two (ID = 12) and three (ID = 13). In what follows, we use the prefix "p" or "n" to denote signed motif types with only positive connections or only negative connections, respectively. **D** Motif intensity significance sequences of all signed motifs (signed three-neuron subnetworks). Each row shows results for a certain stimulus type and each color corresponds to a certain unsigned motif structure. Motif intensity significance for each signed motif is obtained through its intensity Z-score of the empirical network with Signed-pair-preserving model as the reference. The overall sequences are highly similar across visual stimulus classes, with six types of over-represented motifs (pFFLb motifs) and three under-represented motifs preserved. It is worth noting that the above-mentioned nine significant motifs only contain positive connections.

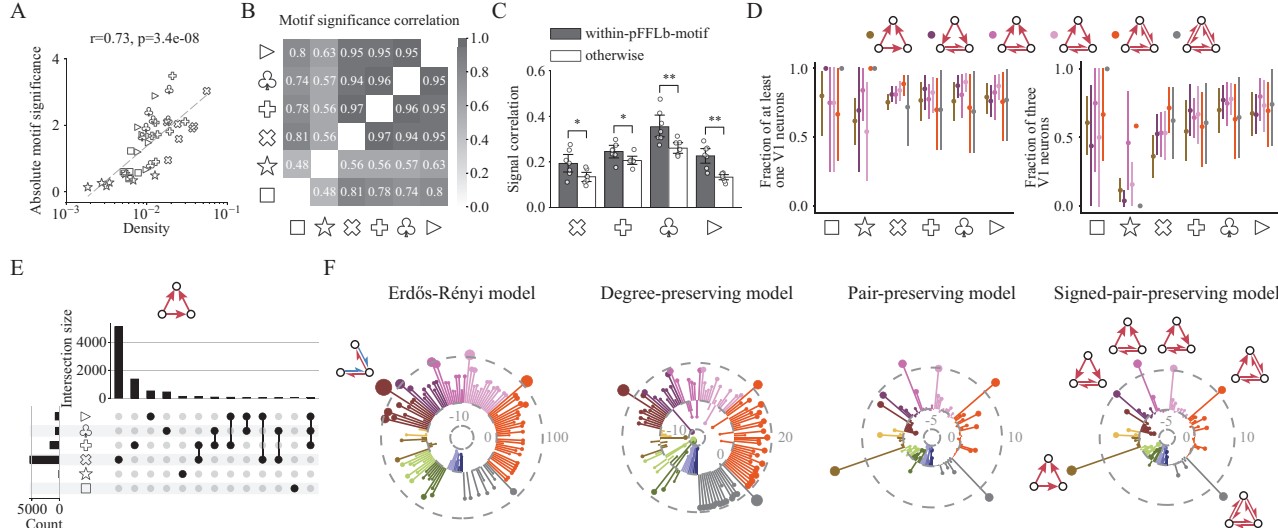

**Fig. 3 | Same motifs and similar patterns are organized from different neurons.**
**A** Averaged absolute motif significance (intensity Z-score) across all signed motifs against network density. Similar to within-area fraction and clustering coefficient (Fig. 1F), there is also a logarithmic relationship between motif significance and density (Wald test, two-sided). **B** Pairwise correlation of normalized motif intensity distribution for six visual stimuli. High correlation proves the similar motif presence during different types of stimuli. **C** Signal correlation during four visual stimuli (except for resting state and flash) for within-pFFLb-motif connections and other connected neuron pairs. *$p < 0.05$, **$p < 0.01$, rank-sum test, one-sided. Error bars represent 95% confidence interval, $n = 7$ mice. **D** Fraction of motifs with at least one V1 neuron or all three V1 neurons during all visual stimuli for six over-represented positive-feedforward-loop-based (pFFLb) motifs. The variation in regional composition indicates the change of constituent neurons for pFFLb motifs across different visual stimuli. Error bars represent 95% confidence interval.

**E** Intersections of unique motif sets for motif ID = p6 during six types of stimuli. Horizontal bar plot shows the number of signed motif ID = p6 for each type of stimulus while vertical bar plot displays the size of each intersection set. A unique motif is defined as a certain signed motif with three specific neurons, and intersections with less than 20 elements are removed for brevity (see Supplementary Fig. 12 for the complete results on all pFFLb motifs). A large number of unique motifs appear only during one type of stimuli, demonstrating that even though functional motifs are preserved across visual stimuli, constituent neurons are changing. **F** Multiple motif intensity significance sequences were obtained through four different reference models for natural movies as the representative stimulus type. Motif order (clockwise) and color are consistent with (**D**) and Fig. 2D, and connectivity pattern is shown only once for each type of most significant signed motif for brevity.

well as acceleration and delay of response[78]. Here we denote all six of the over-represented signed motifs as positive-feedforward-loop-based (pFFLb) motifs.

In addition to the three highly under-represented motifs (ID = p4, p5, p8), the set of under-represented motif patterns consists largely of 'unclosed' pFFLb motifs, suggesting that neurons tend to form pairwise-connected triplets. This under-representation of unclosed motifs appears to be more pronounced with higher network density. Interestingly, the motifs' average absolute intensity Z-scores (deviation from the frequency expected by chance) increases significantly with increasing network density (Fig. 3A; Wald test, two-sided). With more connections, the empirical functional network deviates more strongly from randomized surrogate networks, highlighting the fundamental non-randomness of local functional connectivity.

Note that it is important to examine the whole significance distribution of motifs instead of focusing on merely the most striking ones. Varying the threshold on significant motifs naturally changes their total count, but importantly, it does not substantially change the relationship between the functional connectivity patterns observed for the different stimuli (Supplementary Fig. 7B). Our control analysis has led us to the conclusion that the presence of non-random local topology of functional networks is contingent upon the density of the network (Fig. 3A), which is, in turn, modulated by sensory input (Fig. 1E). Meanwhile, the same sets of two- and three-neuron connectivity motifs are over- and under-represented for all 6 visual stimuli.

Our analysis of the connectivity motifs relies on comparing motif frequencies in the observed network to those reference networks that are similar in some way to the observed network but are otherwise randomized. Notably, there are many different ways to define random networks by preserving certain network properties. For example, the

commonly used Erdős–Rényi reference network preserves the network density. Other reference models can preserve the degree distribution, the neuron pair distribution, or the signed neuron pair distribution (Fig. 2A; Methods). To understand how our motif analysis depends on the choice of reference model, we computed the three-neuron motif intensities for the natural movie stimulus for 4 different reference models with increasingly more preserved network properties (from left to right in Fig. 3F). As a result, the overall significance level for all motif types roughly decreases in the same order, and the most strictly conserved reference model, the signed pair-preserving reference model, is better in identifying the small subset of motifs for which the observed network is most truly non-random. Therefore, we use the signed pair-preserving reference model as our default reference model throughout this study.

To test the robustness of our results, we separately calculated the motif distribution patterns on distinct halves of the trials from our natural scene stimulus data. The high consistency between the motif distributions on the two data splits (Supplementary Fig. 8) suggests that noise within the dataset is unlikely to be a key factor in our results. Furthermore, pFFLb motifs consistently emerge as the most salient motifs across different stimuli at various significance levels (Supplementary Fig. 9), with increased CCG window size (Supplementary Fig. 10) and when considering potential joint modulation from unobserved neurons (Supplementary Fig. 11). Taken together, pFFLb motifs seem to be reliably the most significant patterns among three-neuron subgraphs.

### Properties of over-represented three-neuron motifs

While the three-neuron functional motifs that were most over- or under-represented were the same for all stimuli (Fig. 3B), those motifs were composed of different neurons for different stimulus types

(Fig. 3D, E). Specifically, while most pFFLb motifs contained at least one neuron from V1, the fraction containing all 3 neurons within V1 varied substantially as the stimulus type changed. (Fig. 3D).

To further quantify the extent to which the same neurons constitute pFFLb motifs across stimuli, we computed the number of pFFLb motifs sharing exactly the same neurons for all pairs of stimuli (intersection sizes), as shown in Fig. 3E for one example over-represented pFFLb motif (ID = p6; results for all of the over-represented pFFLb motifs are shown in Supplementary Fig. 12). We also compared motif overlapping detected using half the trials of the same stimulus type and using different stimuli (Supplementary Fig. 8C). Notably, pFFLb motifs were more likely to be composed of the same neurons during the same stimulus type than for different types, indicating that the identities of the neurons within the over-represented motifs change as the stimulus type changes.

Taken together, these results and those in Fig. 2D, indicate that, as the stimulus type changes, the same three-neuron motifs are over- or under-represented in the cortical networks, but the identities of the neurons within those motifs change. This suggests that these specific motifs might have strong functional importance for the cortical microcircuit because even as different stimuli dynamically alter the functional connectivity, they do so in a way that preserves these motif patterns. To further investigate this question, we analyzed the tuning similarity of these motifs. As a control, we compared the tuning profiles of connected neurons that are not within the same pFFLb motifs. We found the signal correlation between pairs of connected neurons within the same pFFLb motif were significantly higher than those not within the same pFFLb motif (Fig. 3C; rank-sum test, one-sided), and neurons within the same pFFLb motif are spatially closer to each other than otherwise (Supplementary Fig. 7A, right; rank-sum test, one-sided). Furthermore, neuron pairs within pFFLb motifs show stronger functional connection strengths and higher CCG Z-scores than other connected neuron pairs (Supplementary Fig. 7A; rank-sum test, one-sided). Thus, the pFFLb motifs tend to consist of functionally-similar neurons. These observations are consistent with previous reports of synaptic connectivity patterns in visual cortex[1].

Overall, these analyses indicate that neuron pairs within the over-represented pFFLb motifs tend to be spatially near each other, and to have higher functional similarity compared to other pairs. Coupled with the fact that these pFFLb motifs are preserved across stimuli, this highlights the potential functional importance of these motifs within the cortical microcircuit.

## Spatial and functional organization of network modules depends on the stimulus

The adoption of network analysis in the investigation of neural mechanisms underlying visual processing offers more than just the exploration of elementary information processing components (e.g., three-neuron motifs). It opens the door to a comprehensive examination of how extensive neural populations, including modular structure, engage in interactions. Those modules[79,80] are thought to impart added robustness[81], efficiency[82], and functional specialization[55] to networks. We thus sought to identify modules within our networks, and to determine how their properties depend on the stimulus presented to the animal.

To achieve this goal, we revised the Louvain method[83] to optimize the Modularity estimation from previous work[84,85] so as to take into account the signs of the connections in our networks: this modified Louvain method searched for sets of modules with most positive connections inside the same module and negative connections between different modules (see Methods). Thus, the method identifies sets of modules whose neurons are internally correlated and externally anti- or un-correlated. This greedy optimization yields the groupings of neurons into modules by maximizing the score of modified Modularity (see Methods). For comparison, we also identified modules with the original Modularity algorithm that does not take into account the edge

signs[81]. Although our adapted method revealed results qualitatively similar to the original one (Supplementary Fig. 13), the identified module size using our method is relatively smaller, suggesting a finer scale module detection with our method.

Unless otherwise stated, in the rest of this paper Modularity means the modified Modularity for signed module detection. By maximizing the two-dimensional Modularity difference map whose dimensions correspond to resolution parameters for positive and negative links, we determined the optimal resolution parameters that control the scale of identified modular structure so that the empirical network deviates most from the null model[15,86] (see Methods; Supplementary Fig. 14A). In light of the potential limitations posed by a fixed resolution parameter, we analyzed multi-resolution module partitioning and found consistent results (Supplementary Fig. 14B–D).

After identifying the best parameters, we applied our module detection algorithm to the observed functional networks from each stimulus type, and compared the results between stimuli. During gratings and natural stimuli, functional networks tend to exhibit stronger modular structures, characterized by larger deviations in Modularity from expectation (Fig. 4A, bottom Z-score, $52.04 \pm 5.16$; mean $\pm$ sem, $n = 4$ stimulus types over 7 mice). On the contrary, the networks obtained from flashes and in the resting state were less modular ($4.18 \pm 4.14$; mean $\pm$ sem, $n = 2$ stimulus types over 7 mice; $p = 5.23 \times 10^{-7}$, rank-sum test, one-sided).

Anatomically parcellated brain regions are thought to work as natural modules with specialized functions[87–90]. We thus wanted to understand how our functionally-defined modules relate to the anatomically-defined brain regions. To achieve this goal, we analyzed the identified functional modules to understand the extent to which their spatial organization coincided with the anatomically-defined brain regions, and the extent to which that depended on the stimulus. To do this, we computed three measures from the modules for each stimulus (Fig. 4B, left). First, the coverage quantifies the maximum extent to which a given module covers all of the neurons in any single brain region. Second, the purity quantifies the maximum extent to which a given module is contained within any single brain region. These two quantities are computed for each module, and the results in Fig. 4C show their weighted average (averaged over modules, weighted by module size). A more detailed module-by-module analysis is presented in Fig. 5, below. Finally, the adjusted rand index (ARI) quantifies the similarity between how the modules partition the set of neurons, and how the brain regions partition the set of neurons. Intuitively, these measures revealed the properties of modular structure from different perspectives: high coverage means at least one visual area is covered by the module, high purity means that a module consists of neurons from the same visual area, and high ARI means the overall module partitioning highly resembles the areal organization.

These three measures all show variation in the module organization for different stimuli (Fig. 4C). One would usually assume that long-range and inter-areal connections are needed for more demanding tasks[91,92]. However, for natural images and movies, the modules have a higher propensity to cover only a subset of a brain region than for drifting and static grating stimuli. This is reflected by lower coverage scores and higher purity scores for the natural image and movie networks than for the drifting and static grating networks (Fig. 4C; $p = 0.0029$, $p = 0.0002$, rank-sum test, one-sided). Therefore, it appears that enhanced within-areal connections play a beneficial role in visual computation which could be more important in the processing of natural stimuli.

It is important to note that with increasing module size, coverage tends to increase while purity tends to decrease, and the module size does depend on the stimulus (Fig. 5B). As a result, it is important to ask whether the variations in coverage and purity with visual stimuli could be explained simply by stimulus dependence of module size. To address this question, we compared module purity and coverage to

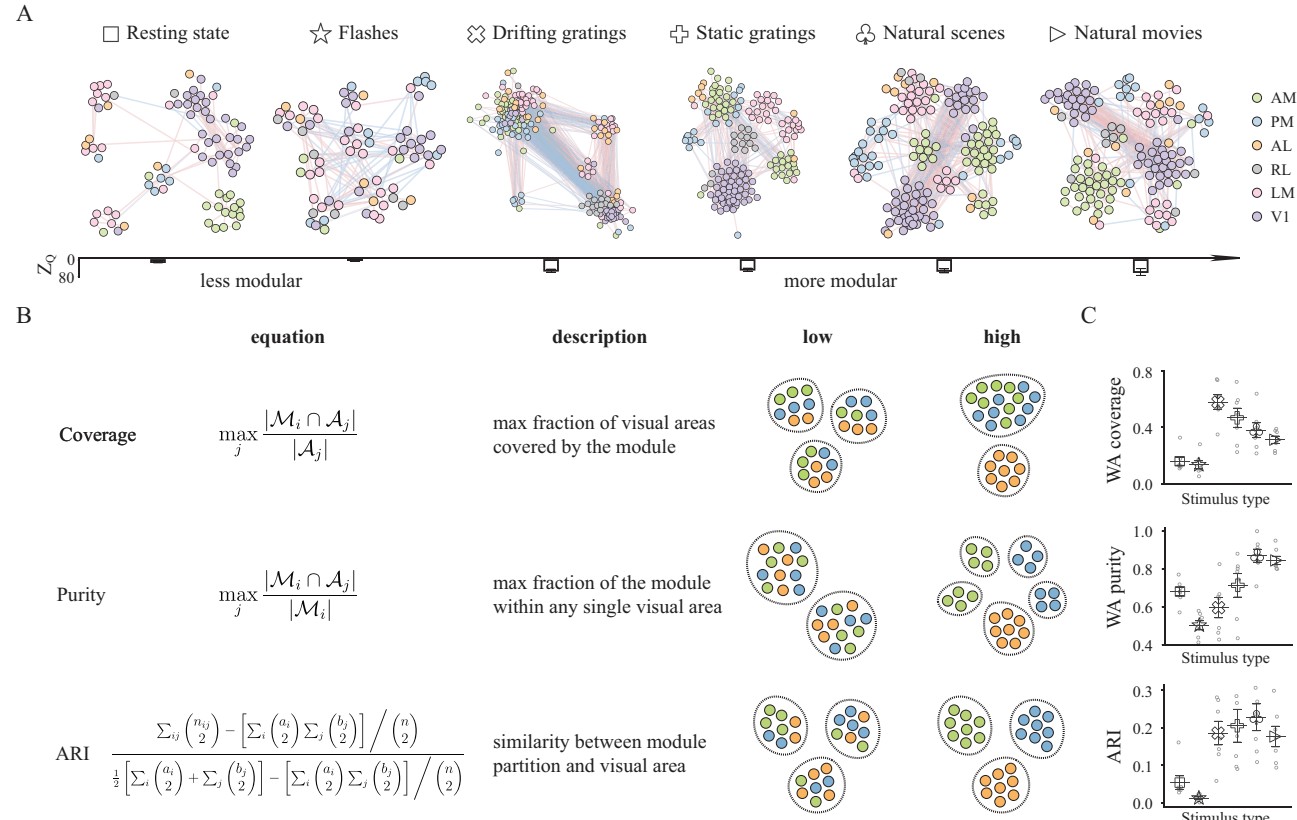

**Fig. 4 | Distinct modular structures during different types of visual stimuli.**
**A** (top) Topological structure of functional connectivity of a mouse during six types
of visual stimuli with neurons colored by area. The color of each connection shows
its sign with red denoting positive connection and blue representing negative
correlation. The community partition is obtained through modified Modularity for
signed networks (see Methods). (bottom) We computed the $Z$-score of Modularity
with Signed-pair-preserving model as the reference to show the degree to which
functional network has a modular structure. Networks during gratings and natural
stimuli show significant modular structure. Error bars represent standard error of
the mean and are displayed with mean, $n = 7$ mice. **B** We used three measures to
reveal the modular structure regarding visual area from different perspectives.
Coverage and purity are module-level measures, where the former marks the
degree to which the module covers any visual area, while the latter measures the
degree to which all neurons in the module are from the same visual area. We

computed the average coverage and purity weighted by module size to show the
overall properties of the whole functional network (see Methods). Adjusted Rand
Index (ARI), a network-level measure, was also used to quantify the difference
between module partition and visual areal organization. The weighted average
(WA) coverage is 0.375 and 1 (ranges from 0 to 1), WA purity is 0.333 and 1 (ranges
from 0 to 1) and ARI is −0.03 and 1 (ranges from −0.5 to 1) for the corresponding two
toy examples visualizing the "low" and "high" cases for the measure. **C** WA cover-
age, WA purity and ARI during six visual stimuli. In general, there tend to be fewer
and larger modules with higher coverage during grating stimuli, whereas we usually
find more and smaller modules with higher purity during natural stimuli. As a result,
ARI is lower during resting state and flash while higher during grating and natural
stimuli. Error bars represent standard error of the mean and are displayed with
mean, $n = 7$ mice.

module size (Supplementary Fig. 15). Consistently across module sizes,
the modules in the flashes stimulus and resting state networks had
lower coverage than did the networks for the other stimuli. The net-
work from the flashes stimulus also had consistently lower purity.

To further probe the relationships between module size and
coverage or purity, we analyzed the number of modules obtained for
each stimulus that were above a given threshold of module size,
threshold of module coverage, or threshold of module purity.
Repeating this for many threshold values (Fig. 5B), we found that
natural images and movies had the largest numbers of high-purity
modules even though their numbers of small modules were not
appreciably different from the other stimuli. These findings emphasize
that the stimulus-dependent module properties we report in Fig. 4C
cannot entirely be attributed to stimulus-dependent module sizes.

In general, the similarity between functional module partitioning
and the anatomical areal organization is higher during gratings and
natural stimuli and lower during resting state and flashes, suggesting
that functional connectivity tends to be more constrained by
the anatomical structure and more spatially compact during complex
and natural stimuli. This is reflected in the lower ARI values for the

flashes and spontaneous activity (Fig. 4C; $p = 4.0 \times 10^{-7}$, rank-sum test,
one-sided).

Consistent with previous work[69,93], functional motifs seem to be
more pronounced in more modular networks (Supplementary
Fig. 16F), suggesting their shared organizational principles. Similar to
motifs, we also tested the functional similarity of nodes within and
across modules by measuring the signal correlations of connected
neuron pairs. Neuron pairs within the same module had higher signal
correlations than did neuron pairs that were not (Fig. 5A), and the
probability of any two connected neurons being in the same module
also increases with signal correlation (Supplementary Fig. 16A;
Cochran–Armitage test, two-sided). These findings were consistent for
the 4 visual stimuli for which the signal correlations are well-defined,
and were consistent across brain regions with modules assigned to the
brain region from which most of their neurons came (Fig. 5D, E).
Functional interactions tend to be found between neurons with more
similar receptive fields across most scenarios concerning areal and
modular structures, although how strongly interactions depend on
receptive field similarity varies (Fig. 5F). This shows that functional
modules are finer-scale partitions, displaying both the similarity and

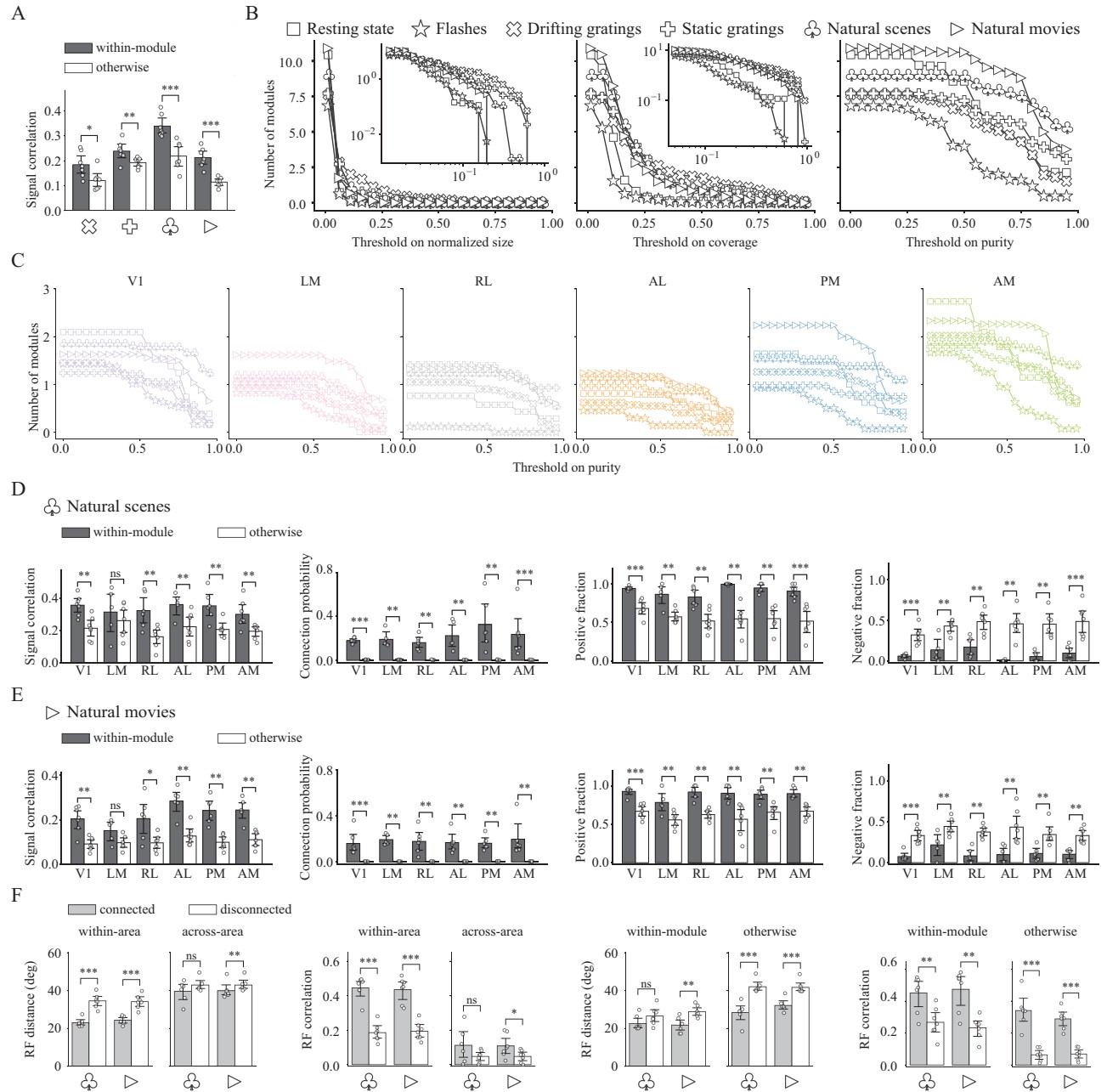

**Fig. 5 | Stronger segregation during natural stimuli. A** Signal correlation for within-module and otherwise connections. Note that some neurons may not belong to any module, otherwise connections can involve multiple scenarios. *$p < 0.05$, **$p < 0.01$, ***$p < 0.001$, rank-sum test, one-sided, $n = 7$ mice. Error bars represent 95% confidence interval. Connected neurons partitioned into the same module tend to have higher signal correlations than the rest connected neurons, demonstrating our module partition provides insight into not only the connectivity pattern but also functional similarity to some extent. **B** Number of modules with normalized size, coverage or purity higher than the threshold. Normalized size is the size of module normalized by the total number of neurons in the network, insets show the plots on a log-log scale. **C** Number of modules with purity higher

than the threshold for each visual area during all visual stimuli. **D**, **E** Properties of the modular structure during natural scene and natural movie presentations. We examined four different aspects of the case where neurons from a single visual area are divided into multiple modules (in which they are the dominant area), with signal correlation indicating the functional similarity along with connection probability, positive and negative fractions demonstrating the validity of our module partitioning algorithm. ns $p > 0.05$, *$p < 0.05$, **$p < 0.01$, ***$p < 0.001$, rank-sum test, one-sided, $n = 7$ mice. Error bars represent 95% confidence interval. **F** Distance in degree between centers and overall correlation of receptive fields for connected and disconnected neuron. ns $p > 0.05$, *$p < 0.05$, **$p < 0.01$, ***$p < 0.001$, rank-sum test, one-sided, $n = 7$ mice. Error bars represent 95% confidence interval.

dissimilarity between anatomical and functional parcellation. These observations emphasize that the modular structure promotes functional specialization[55].

Our analyses of the modular organization of the functional networks reveal that the modules tend to contain neurons with similar stimulus tuning, and that their spatial organization and alignment with anatomical brain regions depend on the stimulus type presented to the

animal. This emphasizes that a functional module is not strictly the same as an anatomical brain region: the relationship between these concepts depends on the stimulus-defined context.

## Discussion
We studied the topology of micro-scale functional networks measured with single-neuron spiking activity in the mouse visual cortex. These

data were collected while the mice were exposed to different types of visual stimuli, and we separately analyzed the functional networks observed in the responses to each stimulus type. Thus generated functional networks could differ from the underlying anatomical connectivity, and this disparity warrants caution when interpreting the connectivity graphs. However, our science question, concerning stimulus-dependent interactions between neurons cannot be answered with the standard anatomical connection methods. For this reason, we used functional connectivity measures for this study.

We found that functional networks display stimulus-dependent network properties such as varying density, clustering coefficient and fraction of positive connections. Furthermore, we provide evidence that the distribution of low-order connectivity patterns (motifs with 2 or 3 nodes) remains stable, characterized by over-representation of a specific group of 3-neuron motifs, pFFLb motifs. This over-representation was preserved across the wide range of stimuli we investigated. Notably, while these motifs were over-represented in all cases, the constituent neurons within those motifs changed. Finally, we observed that the module-level network architecture depends significantly on the stimulus type.

The consistent over-representation of pFFLb motifs suggests that they are key information-processing components of neural circuits. While these motifs were over-represented for all stimuli, the identity and areal distribution of neurons constituting the pFFLb motifs differed between stimuli. Thus it is the three-neuron patterns rather than the triadic interactions of specific neurons that are preserved. This observation suggests that an important computational role might arise at the motif level[94–98], where neurons can dynamically reorganize to form these relevant structures. These local computations, organized by motifs, could remain robust to changes like the loss of individual neurons because other neurons could be recruited into the motifs to replace any that are lost. For this reason, motif-level computational organization could provide substantial robustness to cortical computation.

The abundance of FFL motif has been observed in numerous types of networks including gene regulatory networks[70], transportation networks[99], engineered networks[67] and neuronal networks[1,67,99]. Motif ID = p6 (pFFL) is proven to be a sign-sensitive filter that responds only to persistent stimuli in transcriptional regulation networks[70] and multi-input FFL generalization is found to store memory as well as reject transient input fluctuations in neuronal networks[52]. For anatomical networks of neurons, there have been modeling work showing that certain pFFLb motifs (p9, p10) can function as long-term memories of the input, thus playing an important role in many cognitive tasks[78]. Therefore, the pFFLb motifs that we found to be consistently over-represented in cortical functional networks may have important functional roles in cortical computation.

On the global scale, however, natural stimuli tend to drive networks into more modular structures with stronger segregation and stronger agreement between structural and functional parcellation. Although the extensive inter-areal connections and modules evoked by gratings could result from the intrinsic spatial distribution of the grating stimuli, the possibility of influence from oscillatory patterns (Supplementary Fig. 17) has been ruled out. In comparison, the functional modules with more spatially segregated structures evoked by natural stimuli could play an important role in the processing of the visual input. It is noteworthy that, major functional modules observed when the animal was viewing natural scenes and movies are highly overlapping at all spatial scales with almost the same neurons (Supplementary Fig. 14B). These probably arise from the common subtasks required by the visual processing of similar stimuli. One advantage of having shared modular components is that it allows a faster adaptation and possibly a lower switching cost of functional networks to various tasks[93]. This reduction in functional reorganization costs could be especially important given our observation that different neurons are

organized into the over-represented pFFLb motifs in the presence of different stimuli.

Anatomical structure has been known to stay relatively stable given different types of sensory inputs[100]. In comparison, functional connectivity changes in such a fast and dramatic manner that some even try to model its temporality within a single trial[101]. There are various reasons for this rapid functional adaptation, which could be a change of task[36–39], perceptual states[102], visual stimuli[24,25,103], etc. However, most prior studies were either restricted to the primary visual cortex, or to voxel-level recordings obtained through fMRI, or both. It thus remains unclear whether the functional interactions on a single-neuron scale across multiple cortical regions are also dynamically adapted to the visual inputs. One major aim of our study was to fill this knowledge gap. By studying interneuronal functional connections, our work could help improve our understanding of neuron-to-neuron connections (e.g., at the synaptic scale). In contrast, studies of voxel-scale functional connectivity based on fMRI data might be less informative about these finer-scale interactions.

One of the major challenges of studying functional connectivity lies in the existence of negative correlations and anti-correlated patterns: there is a lack of strong theoretical tools for analyzing networks with both positive and negative edges[104]. While many studies on functional connectivity disregard inhibition and only focus on positive connections for simplicity, inhibitory connections that could display as negative connections play a crucial functional role in visual processing[105,106]. This highlights the need for a network analysis framework that possesses the ability to handle both positive and negative edges. We address this problem by adopting and modifying motif and module detection methods for signed networks, and used these methods to investigate how (signed) functional networks vary on local and global scales. The inclusion of edge sign in motif analysis enables us to further distinguish motifs, since the same unsigned connectivity could correspond to different functions depending on the edge signs[70]. The definition of functional modules can be subjective regarding whether to keep negative connections inside or between modules. Nonetheless, we showed that even though qualitatively similar conclusions can be drawn without distinguishing functional inhibition from excitation (Supplementary Fig. 13), ignoring edge sign (as in prior studies) could lead to a less detailed understanding of the exact pattern of functional segregation and specialization.

While our detailed methods thus provide a more comprehensive analysis of how functional connectivity flexibly adapts to the statistics of visual input, we recognize some limitations to our analyses. First, we do not distinguish neurons according to their cell types. This limits our ability to relate our functional connectivity results to the growing literature on microcircuit architectures. In addition, due to incomplete recording, we observe only a subset of the neurons in each brain area. Correlation-based network inference can potentially lead to false direct-edge identification via high-order connections or joint modulation from latent variables (unobserved units). These limitations are inherent to single-unit neural activity-based construction of functional connectivity, and not just to our study. Nevertheless, we do not believe that these common limitations constitute serious flaws in our analysis. First, we are not trying to find functional networks that topologically resemble the anatomical network, in which case the incomplete recording issue would be quite detrimental, or effective/causal networks that reflect direct influence between neurons where latent variables introduce a significant bias to the identified connectivity. Instead, our focus is on the alteration of inter-neuronal interaction patterns to different visual stimuli measured by correlated activities. These interactions can be identified even in incomplete recordings. Second, incomplete observation might explain the presence of non-conforming edges in some of our analyses, such as the presence of both positive and negative edges emanating from a single neuron. At first glance, these neurons are at odds with Dale's principle, which

suggests that such bivalent neurons are very uncommon in the neocortex. However, given the incomplete recordings, there could be unobserved inhibitory neurons that mediate the effective inhibitory impact of an excitatory neuron on some other neurons in the circuit. Hence, the potential incorporation of indirect interactions within functional connections enables the examination of effective interaction between neurons, even with unrecorded units. Third, despite the demonstrated efficacy of various methods to correct for latent variables in the construction of causal graphs[107–110], the applicability of these models in real-world analyses, particularly in the absence of ground truth connectivity, remains uncertain. It is imperative to underscore that, our study primarily focuses on functional connectivity, which reveals correlations instead of causality between neural activities under different stimulus conditions[111]. For this reason, without presuming direct mechanistic interactions, jitter-corrected-CCG-based functional connectivity that has been proven effective in capturing the direction of information flow[30,31] is not significantly impacted by latent variables.

Additional limitations of our study arise from experimental constraints and the nature of the Neuropixels dataset collected from extracellular electrophysiology probes. Kilosort2 was used to identify spike times and assign spikes to individuals[112], however, no current spike sorting algorithm can ensure a completely accurate assignment of observed spikes to individual neurons. This means certain nodes in our network could correspond to more than one neuron, or that there could be multiple nodes corresponding to the same neuron[113]. Finally, the limited set of visual stimuli used in our experiments could introduce bias into our analysis since we do not have multiple different sets of stimuli within the same stimulus type. This limitation prevents us from comparing the functional connectivity driven by distinct stimuli within the same category (i.e., more different clips of natural movies). On the other hand, we use a relatively wide range of natural image and natural movie stimuli, and sampled multiple stimulus types of varying complexity. While these laboratory conditions are much more controlled than natural viewing conditions, we nevertheless have determined functional connectivity under a wide range of stimulus conditions.

On the timescale of sensory processing, neuronal networks have relatively fixed anatomical connectivity. Their functional connectivity, however, can and does vary quite substantially. Our work revealed striking patterns to this functional reorganization. These patterns suggest potentially important principles governing cortical computation, such as the dynamical organization of groups of neurons into feedforward loop motifs, and the adjustment of network modularity based on stimulus complexity. Beyond their relevance for basic neuroscience, these findings may provide guidance for how to engineer dynamically robust information processing systems.

## Methods
### Dataset
We analyzed the Neuropixels dataset from Allen Institute[30]. The Neuropixels project uses high-density extracellular electrophysiology probes to record spikes from multiple regions in the mouse brain. Data used to construct functional networks are recordings of the neural activity by 6 Neuropixels probes in 6 visual cortical areas (V1, LM, RL, AL, PM, AM) from 7 mice while the mice passively viewed a visual stimulus set that contains 6 types of visual stimuli with multiple repeats: gray screen (simulation for resting-state activity), flashes, drifting gratings, static gratings, natural scenes and natural movies. On average, there are $668 \pm 131$ units simultaneously recorded for each mouse. In order to make a fair comparison across different visual stimuli, only neurons with a firing rate of at least 2 Hz during all stimuli are included in our analysis, thus the number of neurons (size of the functional network) is the same for each mouse given different stimuli. As a result, there are $176 \pm 44$ units on average for each mouse.

### Cross-correlogram and significant functional connection
Functional connectivity is measured through Cross-correlograms (CCGs)[114]. For each stimulus type, the average CCGs across all stimulus presentations is calculated. In order to focus on the change in connectivity driven by different stimulus types, we dismissed stimulus conditions and used all presentations as trials. CCG for lagged correlation from neurons $A$ to $B$ is defined as

$$CCG_{AB}(\tau) = \frac{\frac{1}{M}\sum_{i=1}^{M}\sum_{t=1}^{N}x_A^i(t)x_B^i(t+\tau)}{\theta(\tau)\sqrt{\lambda_A\lambda_B}} \tag{1}$$

where $M$ is the number of trials, $N$ is the number of time bins, $x_A^i$ and $x_B^i$ are the spike trains for neuron $A$ and neuron $B$, $\tau \geq 0$ is the time lag between the spike trains, $\theta(\tau) = N - \tau$ represents a triangle function that corrects for the overlap time bins, $\lambda_A$ and $\lambda_B$ are the mean firing rates for the two neurons. It is worth noting that we only allow for non-negative time lag for the sake of bidirectional connections. We used the jitter correction method to remove slow temporal correlations[115]. The spike trains are split into short jitter windows with equal size and the spike-jitter method perturbs the spike timing while maintaining the number of spikes within each jitter window to disrupt fast-time correlations. Jitter correction has been widely used to eliminate the shared trial-to-trial variability on a slower time scale[116]. The jitter-corrected CCG is obtained as the difference between CCGs of the original and jittered spike trains

$$CCG_{jc} = CCG - CCG_{jittered} \tag{2}$$

The size of the jitter window is defined as 25 ms. Apart from using 'sharp peaks' to define significant functional connections, we also included 'sharp intervals' to take into consideration the polysynaptic connections between neuron pairs with potentially multiple time lags. Specifically, for a given duration $D \in [1, \tau_{max} + 1]$, where $\tau_{max} = 12$ ms similar to the 13 ms window in previous work[31], the set of moving average CCG is obtained by

$$C(D) = \left\{ \frac{1}{D}\sum_{\tau=t_1}^{t_1+D-1} CCG_{jc}(\tau), t_1 \in [0, T-D+1] \right\} \tag{3}$$

where $T$ is the total length of spike trains. Therefore, there is a positive connection if

$$\frac{1}{D}\sum_{\tau=t_0}^{t_0+D-1} CCG_{jc}(\tau) > \mu_{C(D)} + n\sigma_{C(D)} \tag{4}$$

and a negative connection if

$$\frac{1}{D}\sum_{\tau=t_0}^{t_0+D-1} CCG_{jc}(\tau) < \mu_{C(D)} - n\sigma_{C(D)} \tag{5}$$

where $t_0 \in [0, \tau_{max} - D + 1]$ is the starting time lag of the 'sharp peak/interval', $\mu_{C(D)}$, $\sigma_{C(D)}$ are the mean and standard deviation of $C(D)$, $n = 4$ denotes the 4-fold significance level in our experiment. It is straightforward that $D = 1$ indicates a 'sharp peak' while $D > 1$ denotes a 'sharp interval'. If equation (4) or (5) is true on multiple durations $D \in \{D_1, D_2, \cdots, D_s\}, D_1 < D_2 < \cdots < D_s$, we assume the smallest duration $D = D_1$ since it always leads to the highest significance level.

CCG peaks with zero time lag are explained by dynamical relaying mechanisms instead of common drive[117,118], thus are usually treated as bidirectional connections[42]. Considering our CCG method can detect bidirectional connections with non-zero peaks, for connections with zero time lag, we retain the primary direction of communication,

**Table 1 | Reference models**

| Reference model | size & density & weight | degree distribution | pair distribution | signed pair distribution |
|---|---|---|---|---|
| Erdős–Rényi model | ✓ | × | × | × |
| Degree-preserving model | ✓ | ✓ | × | × |
| Pair-preserving model | ✓ | ✓ | ✓ | × |
| Signed-pair-preserving model | ✓ | ✓ | ✓ | ✓ |

defined as the direction with a higher significance level. The alternative direction retains its significance when a non-zero significant peak or interval is observed.

Therefore, each directed connection was characterized by its lag, the duration of the significant interval in the CCG, and its significance value. Lag $\tau$ is the delay between spike trains of source neuron and target neuron. The duration $D$ measures how long the significant peak/interval lasts, and the connection significance signals the Z-score of the 'sharp peak/interval'. Lags $\tau$ of across-area connections are higher than within-area connections (Supplementary Fig. 2C), which is as expected since it takes more time for a signal to travel between areas than within an area.

It has been shown that a sufficiently large number of spikes for both neurons are necessary to make the detection reliable[119], which is directly presented by the CCG. To eliminate the bias brought by insufficient spikes, we used normalized entropy, defined as the Shannon entropy of the CCG divided by the maximum of entropy, to keep only reliable connections where both neurons have enough spikes. We used a threshold of 0.9 on normalized entropy, and most removed connections are found during flashes and are due to the shortage of trials.

## Reference model and signed motif analysis

Since functional networks are constructed as signed networks, signed motif analysis needs to be defined. Similar to unsigned motif detection, to examine the statistical significance of signed n-neuron motifs in the networks, we generated random networks using various reference models as the baseline and conducted a comparative analysis of motif frequency between the empirical network and random networks.

Three types of commonly used reference models are adopted in this work: Erdős–Rényi model, Degree-preserving model and Pair-preserving model. However, they are all defined on unsigned networks. In order to tailor these models for analysis in the context of signed networks, we randomly assigned original edge signs to reference networks randomized using Erdős–Rényi model, Degree-preserving model and Pair-preserving model. Furthermore, we defined the Signed-pair-preserving model by preserving the edge signs for each neuron pair in the Pair-preserving model during shuffling (Fig. 2A). Therefore, surrogate networks generated using all four reference models have the same number of positive/negative connections as the real network.

Table 1 lists the comparison between all four reference models. Erdős–Rényi model randomly shuffles connections while preserving network size, density and weight distribution[77], Degree-preserving model generates random networks while preserving size, density, weight distribution and degree distribution[120], Pair-preserving model randomizes the network while keeping size, density, weight distribution, degree distribution and neuron pair distribution[67] while the Signed-pair-preserving model preserves the signed pair distribution (Fig. 2A) in addition to the first three properties. Furthermore, we also preserved the discretized distribution of anatomical distance between neurons (Supplementary Fig. 5) to test whether spatial distribution has an impact on the findings. However, we used Signed-pair-preserving model for signed motif analysis due to limited data of spatial information. For all analyses including a reference model, we randomly generated 200 surrogate networks.

For two-neuron motif analysis, we adopted Erdős–Rényi model as the reference model and computed the relative count for each type of

two-neuron connection by dividing the count of the empirical network and the average count of surrogate networks. For simplicity, we only focused on three-neuron subnetworks apart from two-neuron subnetworks during motif analysis. We used the Z-score of intensity compared with reference models to determine motif significance[76]. The intensity of a certain motif $M$ is defined as the summation over the intensities of all subgraphs $g$ that have the structure of $M$

$$I(M) = \sum_{g \in M} I(g) \tag{6}$$

where the intensity of a certain subgraph is defined as the geometric mean of all its connection strengths

$$I(g) = \left( \prod_{ij \in l_g} w_{ij} \right)^{\frac{1}{|l_g|}} \tag{7}$$

where $l_g$ denotes the set of connections in $g$ and $w_{ij}$ is the strength of the connection from neuron $i$ to $j$. Then the Z-score of intensity for motif $M$ can be computed as

$$Z_M = \frac{I_M - \langle i_M \rangle}{\sqrt{\langle i_M^2 \rangle - \langle i_M \rangle^2}} \tag{8}$$

where $i_M$ is the total intensity of motif $M$ in one realization of the reference model. To reduce noise from individual mice, we removed outliers of $Z_M$ for each stimulus type from our analysis based on a threshold of 2 standard deviations.

## Signed module detection

The original Modularity used to detect community structure for directed networks[121] is defined as $\hat{Q} = \frac{1}{m} \sum_{ij} [A_{ij} - \frac{k_i^{in} k_j^{out}}{m}] \delta(\sigma_i, \sigma_j)$, where $A$ is the adjacency matrix of the network, $m$ is the number of links, $k^{in}$, $k^{out}$ represent the in-degree and out-degree, respectively. $\delta$ is the Kronecker delta function and $\sigma_i$ denotes the community label that node $i$ is assigned to. In the presence of negative links, we denote $A_{ij}^+ = A_{ij}$ if $A_{ij} \geq 0$ and zero otherwise, $A_{ij}^- = -A_{ij}$ if $A_{ij} \leq 0$ and zero otherwise, so that $A = A^+ - A^-$. In order to cluster nodes towards social balance, frustration metric [122], defined as $\sum_{ij} (\lambda A_{ij}^- - (1-\lambda) A_{ij}^+) \delta(\sigma_i, \sigma_j)$, has been proposed. However, neither is suitable for partitioning signed networks. In this work we adopted modified Modularity for community detection of the signed, weighted and directed CCG network. Modified Modularity of a certain partition $\sigma$ is defined as the weighted combination of the positive and negative parts[84,85]

$$Q(\sigma) = \frac{m^+}{m^+ + m^-} Q^+(\sigma) - \frac{m^-}{m^+ + m^-} Q^-(\sigma) \tag{9}$$

where

$$Q^+(\sigma) = \frac{1}{m^+} \sum_{ij} (A_{ij}^+ - \gamma^+ p_{ij}^+) \delta(\sigma_i, \sigma_j) \tag{10}$$

$$Q^-(\sigma) = \frac{1}{m^-}\sum_{ij}(A_{ij}^- - \gamma^- p_{ij}^-)\delta(\sigma_i, \sigma_j) \tag{11}$$

$\gamma^+$ and $\gamma^-$ are the resolution parameters, $m^+$ and $m^-$ are the number of positive and negative connections, respectively, $p^+$ and $p^-$ are the connection probabilities for positive and negative links, respectively. Here, we take into consideration degree distribution by defining the probabilities as $p_{ij}^{\pm} = \frac{\pm k_i^{out} \pm k_j^{in}}{m^{\pm}}$, where $\pm k_i^{out}$ is the positive/negative out-degree of neuron $i$ and $\pm k_i^{in}$ is the positive/negative in-degree of neuron $j$. Therefore, equation (9) can be rewritten as

$$Q(\sigma) = \frac{1}{m^+ + m^-}\sum_{ij}[A_{ij} - (\gamma^+ p_{ij}^+ - \gamma^- p_{ij}^-)]\delta(\sigma_i, \sigma_j) \tag{12}$$

The Louvain method is a module (community) detection algorithm for partitioning networks into groups of nodes with dense connections within groups and sparse connections between groups[83]. The algorithm uses the original Modularity $\hat{Q}$ as a quality function to optimize the partitioning of the network $\sigma$. The Louvain method operates through a series of iterative steps that merge neighboring modules to maximize the Modularity gain until a locally optimal partition is reached. The algorithm uses a bottom-up approach, starting from single-node modules, and iteratively merges modules to form larger ones. To take into consideration edge signs, we revised the quality function in the Louvain method from original Modularity $\hat{Q}$ to modified Modularity $Q$. Therefore, the modified Louvain method aims to find an optimal partition of nodes such that positive connections are placed within modules while negative connections are between modules.

In order to determine the resolution parameters for module analysis, we obtained a Modularity difference heatmap by varying $\gamma^+$ and $\gamma^-$ and computing the difference between Modularities of empirical and surrogate networks generated by the Signed-pair-preserving model, then looked for the $\gamma^+$ and $\gamma^-$ that maximize the difference[15]. This way we obtained the modular partitioning that is least random. We used the Z-score of Modularity to show how modular a functional network is through comparison with a reference model (Signed-pair-preserving model). The Z-score of Modularity is defined as

$$Z_Q = \frac{Q - \langle q \rangle}{\sqrt{\langle q^2 \rangle - \langle q \rangle^2}} \tag{13}$$

where $q$ is the Modularity in one realization of the reference model. Only modules with a size of at least four neurons are included in subsequent analysis to eliminate the noise influence of isolated single neurons, pairs and triplets. Note that we included connection strength in both motif and module analyses. The CCG peak values represent connection strengths, i.e., we use absolute sum of positive/negative connection weights instead of number of positive/negative connections and the positive/negative degree of a neuron is replaced by total positive/negative connection weights. Unless otherwise stated, $Q$ represents the modified Modularity for signed networks. When visualizing modular structure, the location of each node is determined by applying the Fruchterman-Reingold Layout recursively on the hypergraph and then the subgraph of each community (python package Netgraph).

## Analysis of modular structure

To measure the fundamental properties of modular structure, we used (weighted average) coverage, (weighted average) purity and Adjusted Rand Index (ARI) to show how neurons from different visual areas are clustered together. Coverage, defined as $\max_j \frac{|\mathcal{M}_i \cap \mathcal{A}_j|}{|\mathcal{A}_j|}$ and purity, defined as $\max_j \frac{|\mathcal{M}_i \cap \mathcal{A}_j|}{|\mathcal{M}_i|}$, are module-level metrics, while their weighted averages (WA) with module size as weight are network-level metrics. The WA coverage is

$$\frac{\sum_i \max_j \frac{|\mathcal{M}_i||\mathcal{M}_i \cap \mathcal{A}_j|}{|\mathcal{A}_j|}}{\sum_i |\mathcal{M}_i|} \tag{14}$$

whereas the WA purity is

$$\frac{\sum_i \max_j |\mathcal{M}_i \cap \mathcal{A}_j|}{\sum_i |\mathcal{M}_i|} \tag{15}$$

For each module partition, we also used Adjusted Rand Index (ARI) to measure its similarity to areal organization for each network. Based on the contingency Table 2, ARI is defined as

$$ARI = \frac{\sum_{ij}\binom{n_{ij}}{2} - \left[\sum_i \binom{a_i}{2}\sum_j \binom{b_j}{2}\right]\Big/\binom{n}{2}}{\frac{1}{2}\left[\sum_i \binom{a_i}{2} + \sum_j \binom{b_j}{2}\right] - \left[\sum_i \binom{a_i}{2}\sum_j \binom{b_j}{2}\right]\Big/\binom{n}{2}} \tag{16}$$

## Multi-resolution module partition

To reduce noise in module partition, we focused on the most active neurons that have at least 1 connection during all stimuli when examining how module partition changes with resolution parameters. Since the modified Louvain method is stochastic, 200 independent runs were carried out for partitioning any empirical network. To compare module partitioning results across resolution parameters, we combined multiple partitioning results based on a voting mechanism that keeps frequent modules. In our algorithm, we first recorded the module assignment for each node during each run of the modified Louvain algorithm, initially labeling all nodes as unassigned. For the unassigned nodes, we then updated votes for their modules based on module appearances across all runs. Subsequently, we assigned each unassigned node to the module with the most votes. The last two steps were iterated until all nodes were assigned to a module.

Due to the significantly greater abundance of positive connections compared to negative connections, the parameter $\gamma^+$ exerts a substantially more pronounced effect on the outcomes of module partitioning than $\gamma^-$. Consequently, we limited the range of variation for $\gamma^-$ while placing greater emphasis on the alignment and comparison of module identity with $\gamma^+$ across a broader range of values.

To compare module partitions across multiple resolutions, we assigned module IDs to modules across resolutions based on their hierarchical structure and produced a visualization of the resulting heatmap. To accomplish this, we started from the highest resolution, and traversed through the resolutions in reverse order. For the highest resolution, we assigned each module a unique ID.

**Table 2 | Contingency table for module partition**
$\mathcal{M} = \{\mathcal{M}_1, \mathcal{M}_2, \ldots, \mathcal{M}_r\}$ **and visual areal organization**
$\mathcal{A} = \{\mathcal{A}_1, \mathcal{A}_2, \ldots, \mathcal{A}_s\}$

|  | $\mathcal{A}_1$ | $\mathcal{A}_2$ | ... | $\mathcal{A}_s$ | sums |
|---|---|---|---|---|---|
| $\mathcal{M}_1$ | $n_{11}$ | $n_{12}$ | ... | $n_{1s}$ | $a_1$ |
| $\mathcal{M}_2$ | $n_{21}$ | $n_{22}$ | ... | $n_{2s}$ | $a_2$ |
| ⋮ | ⋮ | ⋮ | ⋱ | ⋮ | ⋮ |
| $\mathcal{M}_r$ | $n_{r1}$ | $n_{r2}$ | ... | $n_{rs}$ | $a_r$ |
| sums | $b_1$ | $b_2$ | ... | $b_s$ |  |

$\mathcal{M}_i$ denotes the set of neurons in the $i$-th module while $\mathcal{A}_j$ represents the set of neurons in the $j$-th visual area. Since we only focus on six cortical areas, thus s = 6. Each entry $n_{ij}$ denotes the number of neurons that are assigned into module $\mathcal{M}_i$ are from visual area $\mathcal{A}_j$: $n_{ij} = |\mathcal{M}_i \cap \mathcal{A}_j|$.

Then for each module under subsequent resolutions, we identified its largest submodule from the previous resolution and determined that its ID should be inherited from the submodule. To achieve this, we considered the modules from the previous resolution and calculated their overlaps with the current module. We selected the submodule(s) with the maximum overlap and retrieved the corresponding ID(s) assigned to it. These ID(s) were assigned to the current module, ensuring consistency and preserving the hierarchical relationship across resolutions.

Once the module IDs were assigned to the modules for all resolutions and stimuli, we sorted the nodes within each area based on their combined similarity across all stimuli to ensure an intuitive visualization. Specifically, we employed a two-opt optimization algorithm to determine an optimal node order that maximizes the similarity between module IDs of adjacent nodes across resolutions. The object is to minimize the hamming distance between 10 adjacent nodes, making neighboring nodes more likely to have similar module IDs across resolutions.

### Statistical analysis

Since module partitioning and the generation of surrogate networks are both stochastic, each analysis involving detection of modular structure or generation of surrogate networks is performed with 200 independent runs. We adopt Cochran–Armitage trend test to assess the association between categorical variables and Chi-squared test for comparing the distribution of categorical variables. Wilcoxon rank-sum test is used to compare population mean of two groups. Kolmogorov–Smirnov test is used to test whether a distribution is the largest among a set of distributions. Benjamini/Hochberg method is used to correct $p$ value for false discovery rate in multiple tests.

We use the modified asymptotic (MA) test for comparing correlations between the firing rate of a neuron and its number of within-area/across-area connections[123]. Since the two correlations in the test involve a common variable (firing rate), a test for overlapping correlations is adopted. Suppose $r_1$, $r_2$ and $r_3$ are the correlations between firing rate and within-area connections, firing rate and across-area connections, and within-area and across-area connections, respectively. The MA method defines the confidence limits $(L, U)$ for the correlation difference $r_1 - r_2$ as

$$L = r_1 - r_2 - \sqrt{(r_1 - l_1)^2 + (u_2 - r_2)^2 - 2\widehat{corr}(r_1, r_2)(r_1 - l_1)(u_2 - r_2)} \tag{17}$$

$$U = r_1 - r_2 + \sqrt{(u_1 - r_1)^2 + (r_2 - l_2)^2 - 2\widehat{corr}(r_1, r_2)(u_1 - r_1)(r_2 - l_2)} \tag{18}$$

where $l, u$ are the confidence limits for single correlation $r$ with a significance level $\alpha$

$$l = r - z_{\frac{\alpha}{2}}\sqrt{\widehat{var}(r)} \tag{19}$$

$$u = r + z_{\frac{\alpha}{2}}\sqrt{\widehat{var}(r)} \tag{20}$$

where $\widehat{var}(r) = \frac{(1 - r^2)^2}{n}$ and $n$ is the sample size. The correlation between the two correlations $\widehat{corr}(r_1, r_2)$ can be estimated by

$$\widehat{corr}(r_1, r_2) = \frac{\widehat{cov}(r_1, r_2)}{\sqrt{\widehat{var}(r_1)\widehat{var}(r_2)}} \tag{21}$$

where covariance $\widehat{cov}(r_1, r_2)$ between the two correlations can be obtained through

$$\widehat{cov}(r_1, r_2) = \frac{(r_3 - \frac{1}{2}r_1 r_2)(1 - r_1^2 - r_2^2 - r_3^2) + r_3^3}{n} \tag{22}$$

Therefore, if $L(r_1, r_2) > 0$, $r_1$ is considered significantly higher than $r_2$ with significance level $\alpha$; on the contrary, if $L(r_2, r_1) > 0$, $r_2$ is considered significantly higher than $r_1$ with significance level $\alpha$.

### Reporting summary

Further information on research design is available in the Nature Portfolio Reporting Summary linked to this article.

## Data availability

All the data analyzed in this manuscript is part of the Allen Brain Observatory introduced in ref. 30. The raw data used to generate main text figures is available for download in Neurodata Without Borders (NWB) format via the AllenSDK. Example Jupyter Notebooks for accessing the data can be found at https://allensdk.readthedocs.io/en/latest/visual_coding_neuropixels.html. Source data for main text figures are provided as a Source Data file. Source data are provided with this paper.

## Code availability

Code for analyses in the manuscript and generation of figures are available from the repository: https://github.com/HChoiLab/functional-network.

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

## Acknowledgements

This work was supported by funding from the Tsinghua-Peking Center for Life Sciences, National Natural Science Foundation of China (92370116) and Tsinghua University Dushi Program (2023Z11DSZ001) to X.J., a Discovery Grant from the Natural Sciences and Engineering Research Council of Canada (RGPIN-2019-06379) and a Canada Research Chair grant to J.Z., and the National Eye Institute of the National Institutes of Health under Award Number R00 EY030840 and a Alfred P. Sloan Research Fellowship in Neuroscience to H.C. The content is solely the responsibility of the authors and does not necessarily represent the official views of the National Institutes of Health. We thank Yu Hu, Joshua H. Siegle, Eric Shea-Brown, and Stefan Mihalas for helpful and insightful comments on the manuscript.

## Author contributions

Conceptualization: D.T., J.Z., X.J., and H.C.; Methodology and software, visualization, and formal analysis: D.T.; Investigation: D.T., J.Z., X.J., and H.C.; Writing: D.T., J.Z., X.J., and H.C.; Supervision: J.Z., X.J., and H.C.

## Competing interests

The authors declare no competing interests.
