## [Peer Review File · Nature Communications]

Stimulus type shapes the topology of cellular functional networks in mouse visual cortexReviewers' comments:

Reviewer #1 (Remarks to the Author):

The paper presents an analysis of high-density electrophysiological recordings in mice using network-theoretical tools. The goal of the study is to assess the dependence of joint activity patterns in pairs, triplets and larger ensembles of neurons on the type of stimuli that animals are presented with in a passive viewing condition. The stimulus types span a large range of simple and complex stimulus types. A goal of the study is to establish polarity-dependent interactions in response to different stimulus types. The study reports the dominance of excitatory, close-knit ensembles (one-to-one connections in pairs and triples, often with reciprocal connections). The findings of the paper can be summarized as:

- * across stimulus types the dominant triplet and doublets motifs are preserved
- * across stimulus types, the identity of neurons participating in joint activity patterns change but the frequency of motif occurrence is largely constant
- * the paper introduces a number of control networks to produce surrogate data, which provides a promising toolset for validating claims about functional network topology

I have a number of major issues with concerning the relevance and conclusions that can be drawn from the results and also a list of more technical and clarification issues.

Major issues

1. The paper identifies doublets and triplets of neurons such that it also classifies the interactions between participating neurons as excitatory or inhibitory. The dominant triplets are exclusively excitatory motifs. The analysis and text suggests that these are lateral interactions between these pairs and triplets. When considering these high-level moments of the response statistics, it is tempting to consider latent variables as a source for such non-trivial joint statistics. While the paper does a nice job contrasting the findings with Erdos Renyi graphs and its variants, the latent-variable account is only very briefly mentioned in the discussion. Anatomical, functional, and statistical considerations all point to the importance of considering that the observed joint activity patterns are explained by joint modulation from regulatory networks within the same brain region or in higher level brain areas. Without considering the joint modulation of the response statistics by not recorded neurons/populations the claims on the importance of ternary interactions between neurons seems to be less convincing. This is particularly relevant from line 184-185 and for the point on the underrepresentation of unclosed motifs (line 192). The incremental approach to fit joint response statistics is obviously severely bounded by the limitations in the amount of data, despite the fact that based on the findings of the paper there is little

to deter the reader from questioning the relevance of the ternary interactions as opposed to interactions between neuron quadruples.

2. Section 2.4 moves from small-scale motifs to modules. The manuscript would benefit from a better integration with the previous part and from a more direct phrasing of its relevance. I believe that the statements such as the one made in line 294 on lower coverage scores of natural images and movies would require some sort of hypothesis on what we would expect from different considerations. The finding presented at this point seems to be well-aligned with coverage simply correlating with pixel correlations, or correlations between low-level features.

3. The paper presents the dependence of the participation of specific neurons on stimulus statistics. This statement is most relevant if there is a quantification of the reliability of the estimates of the motifs identified by the authors. The paper presents a fitting to two halves of the stimulus but I am unsure if the two halves contain identical stimuli and if there is a triplet-by-triplet analysis of the stability of the analysis. Ideally, the control should be done to multiple responses to the same stimulus in order to control for stimulus-induced variances, but I assume that this is prevented by the design of the experimental session (i.e. scheduling of stimuli). Related to this, in line 117 the authors report that 'natural stimuli tend to evoke fewer functional connections than grating stimuli'. Since the analysis assesses total correlations and it seems that full field gratings are characterized by stronger correlations at the pixel level, it is unclear how much the reduction in joint activations is a consequence of smaller pixel correlations.

4. The paper discusses the stimulus dependence of the joint activity patterns. While at some points of the paper it is clearly phrased that it is the dependence on stimulus *statistics* (*type* of stimulus) but ever so often the text makes claims about stimulus dependence of correlations. From the figures it seems that it is always the stimulus type dependence that is investigated but a through clarification of this aspect would be welcome given the importance of this concept. There are studies that attempt to assess how correlations beyond those coming from the similarity/dissimilarity of RF structure dependent on stimulus (characterized as noise correlations) (e.g. apart from the cited Banyai et al paper Lin et al (2015) Neuron; Franke et al (2016) Neuron) and therefore the careful phrasing of the claims would be necessary.

5. Cortical activity in mice notoriously depends on posture, and movements. Controlling for locomotion/movements seems to be especially crucial during the investigation of joint activity patterns: locomotion as a latent variable is prone to co-modulate the activities of multiple neurons. Although the authors point out that an analysis was performed to control for movements (pointing out that the analysis is not shown), the details of such an analysis are pertinent to understand the validity of the results.

Minor:

line 121: The Vinje & Gallant paper seems to investigate the difference between CRF and CRF + NCRF stimulation rather than natural vs non-natural stimuli, as the text suggests

line 123: please define network density more explicitly

line 143-146: the paper reports higher signal correlation for neurons within the same motif: I am not sure if this is simply a direct consequence of motifs being defined through cross-correlations between pairs of neurons.

line 158: this is an example where the term stimulus dependency seems to be confusing. Unless it is indeed stimulus dependency and not stimulus type dependency.

line 169: the reader would benefit from a brief definition of the types of surrogate networks. This is well-motivated technical contribution and it would be great to have an instant grasp on what the different surrogates do

line 209: some support for the interpretation of degree distribution and neuron pair distribution would be beneficial for the reader.

line 223 & 231 & 233: the text should be more specific if it is referring to changes in stimulus identity or changes in stimulus statistics

Reviewer #2 (Remarks to the Author):

Tang et al investigated the functional connectivity among neurons across visual areas during the absence and presentation of visual stimuli. They designed a novel optimization technique for a modularity score with signed weights. This work provides an interesting set of new analyses on an existing dataset that open up additional questions that upon answering I think would greatly improve the scientific impact of the manuscript (listed below). I have only one major concern with the manuscript: the over-interpretation of the difference between functional connectivity for gratings versus natural stimuli – there are substantial differences in the spatial correlations of the two stimulus sets that could explain a large amount of the differences (see point 4).

1) Can the authors briefly describe the running speed control? Do resting-state functionally-connected neurons have similar running speed correlations?

2) Relatedly, how distinct are the functional connections for different stimulus groups? If two neurons are functionally connected based on one stimulus set/resting state, are they more likely to be found to be functionally connected during another stimulus set/resting state? The authors state “While the three-neuron functional connectivity motifs that were most over- or under-represented were the same for 223 all stimuli (Fig. 3B), those motifs were composed of different neurons for different stimuli (Fig. 3D,E).” However, Fig 3D does not show this, and 3E is not understandable to me as someone outside network theory (and is only for the e6 motifs?).

3) Based on the size of the window for the CCG, is it surprising that motif 6 is more common than e.g. 1 or 2? I could imagine that in motif 1 with a large enough window, a connection between the receiving nodes would be identified. It would be useful to show as a function of the window size that the number of FFLs detected does not decrease more than for example the number of 1 and 2 motifs. Also, Figure 3F signed-pair preserving is the reference for Figure 2C – it would be nice if it was introduced in that figure for the reader.

4) The across-area analysis uncovers the potential problem in interpreting the functional correlations in the drifting gratings as significant. The drifting gratings are full-field with the same structure at every position. This may produce more functional correlations that are only related to the structure of the stimulus itself. I don't see how interpreting these functional correlations as different from natural scenes is meaningful beyond this fact of stimulus spatial correlation difference, but perhaps the authors could provide some insight here (Figure 5 continues this point). This statement to me “This emphasizes that a functional module is not strictly the same as an anatomical brain region: the relationship between these concepts depends on the stimulus-defined context” in light of the differences across stimuli seems unavoidable. Also the statement in the discussion about this seems to be an over-interpretation: “On the global scale, however, more complex visual stimuli tend to drive networks into more modular structures with stronger segregation and stronger agreement between structural and functional parcellation. This suggests that functional modules with more spatially segregated structure could be required in more demanding cognitive tasks.”

5) If the authors do want to claim that different stimuli will always create different functional connections, then perhaps a good analysis would be to split the natural scene or natural movie responses in half and compute the functional connections in the two halves. If these are different, then indeed the functional connections are highly dependent on the stimulus. This is likely true if the neural activity is high-dimensional and/or the number of stimuli presented is low.

6) For the across-area connections in the natural scenes/movies, are they more likely for neurons with similar spatial receptive fields? Or is there something else underlying the across-area functional connectivity?

Minor:

What are the definitions of network density and clustering coefficient in Figure 1F? I could not find them in the methods, sorry if I missed it.

Reviewer #3 (Remarks to the Author):

Comments to the authors:

In this manuscript, the authors present differences in functional connectivity for different stimulus modalities in the inter-areal network of neurons. The work is based on open-source datasets from the Allen Brain Observatory and comprises simultaneous recordings from an average of several hundred neurons in 6 different areas of the mouse visual system while the mice passively viewed 6 different stimulus modalities. Functional connectivity was estimated based on surrogate corrected sharp peaks and intervals of cross-correlograms (CCGs). A variety of topological analyses were performed on the resulting networks mainly focusing on stimulus modality dependent differences of the 3-neuron connectivity motifs and the community structure. Thereby, the authors showed that despite of similar distribution of motifs across stimulus modalities, the neurons participating in within those motifs differ significantly between modalities. Furthermore, the degree to which the neurons of the networks could be attributed to modules with strong connectivity within and low connectivity between each other also differed significantly between modalities.

The results of this study are interesting and build on recent advances in the fields of simultaneous recordings, network connectivity of neurons and network neuroscience. Nevertheless, some aspects remain unclear, and some analytical issues could significantly affect the results described. Therefore, I suggest a revision of the following points:

Major comments:

1.) In this study CCGs are used to estimate functional connectivity between neurons for different stimulus modalities. However, it has been known for a long time that the distinctness of peaks and troughs in CCGs strongly depends on the number of spikes/ the firing rate of the pair of neurons. In particular, the lower firing rate strongly determines the strength of the correlation, which can neither be corrected by the geometric mean normalization nor the jitter surrogate subtraction. Furthermore, it is evident from Figure 1 that there are significant differences in the average firing rate between stimulus

modalities in this dataset. Thus, firing rate differences are a confounding factor for the identified differences between stimulus modalities. It would be highly beneficial to the manuscript to add analyses of rate differences between stimulus modalities and to show that the results described cannot be explained by mere differences in firing rate.

2.) Following the previous point, rate differences affect not only the detection of peaks (and troughs) but even more strongly bias statistical detection of significant connections. This could explain a substantial part of the topological differences between modalities described here. Therefore, additional controls would be very helpful to exclude this possibility such as stratification for number of spikes, a repetition of the analyses with only neurons with very high firing rates or a fixed threshold instead of statistics for connection detection.

3.) In the manuscript clear differences are shown between grating (static and moving) and other stimulus modalities. Grating stimuli are well known to cause oscillatory couplings between neurons, which also result in sharp peaks and intervals. It would be important to show that the described differences are not explained by differences in oscillatory connections or if they are, to show these differences.

4.) On page 4 in the lines 168-172 the authors describe the used surrogate networks for normalization and statistical testing of network measures throughout the manuscript. However, in contrast to fMRI-based networks, neurophysiological recordings do not represent the whole network, but are patchy spatial subsamples. Moreover, the number of connections between neurons decreases strongly with distance. Therefore, it is important to resemble the distance dependent connectivity when generating surrogate networks for topological analyses of networks of neurons as described in: (Gerhard et al., 2011; Dann et al., 2016). The strong spatial subsampling could otherwise explain a significant part of the described motif distributions as well as the community structures.

Minor comments:

1.) In the Introduction in line 34 the authors state that: “neurons within the network adapt quickly to different stimuli”. However, the study cited is a PET study in which the activity of neurons is not directly measured, therefore this formulation is misleading.

2.) In the Introduction in the lines 60-61 the authors state that: “Functional connectivity at this finer scale is less well-studied due to challenges in simultaneous recordings from large populations of neurons with high spatial and temporal resolution.” This is not entirely correct, since a number of studies have been published on this subject, which should be cited and the sentence reworded accordingly: (Fujisawa et al., 2008; Yu et al., 2008; Berger et al., 2010; Gerhard et al., 2011; Dann et al., 2016; Nigam et al., 2016; Humphries, 2017).

3.) Following on from point 2) in lines 65-67 (Dann et al., 2016) should be cited.

4.) Following on from point 2) in lines 69-71 (Fujisawa et al., 2008) should be cited.

5.) The order of supplementary figures mentioned in the text is confusing because some later figures are described before earlier figures. I consider reordering the supplementary figures.

- 6.) In the Methods, the used jitter correction method sounds very interesting, but is not properly described in the manuscript. It would be very helpful to add a short description of the method.
- 7.) In the Methods, it is not entirely clear how CCGs with 0 lag peak (or trough) were dealt with.
- 8.) In In the Methods in the lines 463-465 the authors state that: “For each neuron pair, we only keep trials in which spike trains of both neurons have a normalized entropy of at least 0.9.” Discarding trials sounds like it could lead to a biased results.
- 9.) In Figure 3D and in Figure 4C panels would be better understandable with x-axis labels.
- 10.) In Figure 3E the symbols for the different modalities are very hard to recognize.

Berger D, Borgelt C, Louis S, Morrison A, Grün S (2010) Efficient Identification of Assembly Neurons within Massively Parallel Spike Trains. *Comput Intel Neurosc* 2010:439648.

Dann B, Michaels JA, Schaffelhofer S, Scherberger H (2016) Uniting functional network topology and oscillations in the fronto-parietal single unit network of behaving primates. *Elife* 5:e15719 Available at: <http://elifesciences.org/lookup/doi/10.7554/eLife.15719>.

Fujisawa S, Amarasingham A, Harrison MT, Buzsáki G (2008) Behavior-dependent short-term assembly dynamics in the medial prefrontal cortex. *Nat Neurosci* 11:823–833 Available at: <http://www.ncbi.nlm.nih.gov/pmc/articles/PMC2562676/>.

Gerhard F, Pipa G, Lima B, Neuenschwander S, Gerstner W (2011) Extraction of Network Topology From Multi-Electrode Recordings: Is there a Small-World Effect? *Front Comput Neurosci* 5:4 Available at: http://www.frontiersin.org/Computational_Neuroscience/10.3389/fncom.2011.00004/abstract.

Humphries MD (2017) Dynamical networks: Finding, measuring, and tracking neural population activity using network science. *Netw Neurosci* 1:324–338 Available at: http://www.mitpressjournals.org/doi/abs/10.1162/NETN_a_00020.

Nigam S, Shimono M, Ito S, Yeh F-C, Timme N, Myroshnychenko M, Laphs CC, Tosi Z, Hottoway P, Smith WC, Masmanidis SC, Litke AM, Sporns O, Beggs JM (2016) Rich-Club Organization in Effective Connectivity among Cortical Neurons. *The J Neurosci* 36:670–684 Available at: <http://www.jneurosci.org/cgi/doi/10.1523/JNEUROSCI.2177-15.2016>.

Yu S, Huang D, Singer W, Nikolic D (2008) A Small World of Neuronal Synchrony. *Cerebral Cortex* 18:2891–2901 Available at: <http://www.cercor.oxfordjournals.org/cgi/doi/10.1093/cercor/bhn047>.

Rebuttal letter

January 13, 2024

Response to Reviewers' comments:

Reviewer 1 (Remarks to the Author): The paper presents an analysis of high-density electrophysiological recordings in mice using network-theoretical tools. The goal of the study is to assess the dependence of joint activity patterns in pairs, triplets and larger ensembles of neurons on the type of stimuli that awake animals are presented with in a passive viewing condition. The stimulus types span a large range of simple and complex stimulus types. A goal of the study is to establish polarity-dependent interactions in response to different stimulus types. The study reports the dominance of excitatory, close-knit ensembles (one-to-one connections in pairs and triples, often with reciprocal connections). The findings of the paper can be summarized as: * across stimulus types the dominant triplet and doublets motifs are preserved * across stimulus types, the identity of neurons participating in joint activity patterns change but the frequency of motif occurrence is largely constant * the paper introduces a number of control networks to produce surrogate data, which provides a promising toolset for validating claims about functional network topology

I have a number of major issues with concerning the relevance and conclusions that can be drawn from the results and also a list of more technical and clarification issues.

Major issues

1. The paper identifies doublets and triplets of neurons such that it also classifies the interactions between participating neurons as excitatory or inhibitory. The dominant triplets are exclusively excitatory motifs. The analysis and text suggests that these are lateral interactions between these pairs and triplets. When considering these high-level moments of the response statistics, it is tempting to consider latent variables as a source for such non-trivial joint statistics. While the paper does a nice job contrasting the findings with Erdos Renyi graphs and its variants, the latent-variable account is only very briefly mentioned in the discussion. Anatomical, functional, and statistical considerations all point to the importance of considering that the observed joint activity patterns are explained by joint modulation from regulatory networks within the same brain region or in higher level brain areas. Without considering the joint modulation of the response statistics by not recorded neurons/populations the claims on the importance of ternary interactions between neurons seems to be less convincing. This is particularly relevant for line 184-185 and for the point on the underrepresentation of unclosed motifs (line 192). The incremental approach to fit joint response statistics is obviously severely bounded by the limitations in the amount of data, despite the fact that based on the findings of the paper there is little to deter the reader from questioning the relevance of the ternary interactions as opposed to interactions between neuron quadruples.

We extend our gratitude to the reviewer for raising this important concern. In the context of causal inference, the impact of latent variables (such as unobserved units that could be interacting with the recorded units) on effective/causal connections is a recognized challenge. To clarify, we would like to emphasize that our analysis of the functional network mainly addresses (jitter-corrected CCG-based) correlation rather than causality. We have now clarified our text to reflect the essence of the network. We have also modified our interpretation considering the influence of unobserved variables on network properties, even given our analysis of the functional network. To confirm that our main conclusions are not affected by this issue, we performed new analysis (see updated Figure S10) and also added a discussion paragraph in lines 414-438 to go over this issue.

As the reviewer has pointed out, the latent variables, when not properly addressed, may lead to misinterpretations and confounding effects in the inferred causal relationships.

Various methods have been proposed to address this issue, including the incorporation of a hidden thalamic source in dynamic causal modeling [1], Granger causality utilizing sub-network analysis [2], the introduction of both a global temperature variable and an additional latent unit [3], and the modeling hidden confounding as perturbation [4]. These methods have demonstrated enhanced performance in capturing the dynamics or ground truth connectivity in the presence of latent variables.

However, the above methodologies exhibit limitations. Despite their demonstrated efficacy on synthetic data, the applicability of these models in real-world analyses, particularly in the absence of ground truth connectivity, remains uncertain. Furthermore, the technology hurdle of our inability to record all neurons introduces a fundamental challenge in understanding the impact of the latent variables. While these methods are crucial in uncovering effective/causal connections in the presence of latent variables, our study primarily focuses on functional connectivity, which reveals **correlations** instead of causality between neural activities under different stimulus conditions [5]. For this reason, without presuming direct mechanistic interactions, functional connectivity is not significantly impacted by latent variables. Meanwhile, jitter-corrected CCG-based functional connectivity has been demonstrated to be effective in capturing the direction of information flow [6,7]. It is noteworthy to mention that functional motifs (especially triplets) without correction for latent variables have been discovered to reflect the level of consciousness [8,9], accurately predict neural activity [10], encode global topological information [11], and exhibit coupling with anatomical connections [12]. Therefore, while hidden variables may indeed constitute a vital consideration in the context of causal connectivity, our findings regarding functional motifs, representing information flow patterns in triplets, are less affected by the joint modulation from unobserved units. We added an introduction of these previous work in lines 173-176: *Although extensive over-representation of certain network motifs such as feedforward loop is found mostly in anatomical networks [13–16], functional motifs (especially triplets) also have been discovered to reflect the level of consciousness [8,9], accurately predict neural activity [10], encode global topological information [11], and exhibit coupling with anatomical connections [12].*

We also added a paragraph in the discussion section to clarify the distinction of causality and correlation based network and the different impact of hidden variables on the interpretation of the network in lines 414-438 *‘While our detailed methods thus provide a more comprehensive analysis of how functional connectivity flexibly adapts to the statistics of visual input, we recognize some limitations to our analyses. First, we do not distinguish neurons according to their cell types. This limits our ability to relate our functional connectivity results to the growing literature on microcircuit architectures. In addition, due to incomplete recording, we observe only a subset of the neurons in each brain area. Correlation-based network inference can potentially lead to false direct-edge identification via high-order connections or joint modulation from latent variables (unobserved units). These limitations are inherent to single-unit neural activity-based construction of functional connectivity, and not just to our study. Nevertheless, we do not believe that these common limitations constitute serious flaws in our analysis. First, we are not trying to find functional networks that topologically resemble the anatomical network, in which case the incomplete recording issue would be quite detrimental, or effective/causal networks that reflect direct influence between neurons where latent variables introduce a significant bias to the identified connectivity. Instead, our focus is on the adaptation of inter-neuronal interaction patterns to different visual stimuli measured by correlated activities. These interactions can be identified even in incomplete recordings. Second, incomplete observation might explain the presence of nonconforming edges in some of our analyses, such as the presence of both positive and negative edges emanating from a single neuron. At first glance, these neurons are at odds with Dale’s principle, which suggests that such bivalent neurons are very uncommon in the neocortex. However, given the incomplete recordings, there could be unobserved inhibitory neurons that mediate the effective inhibitory impact of an excitatory neuron on some other neurons in the circuit. Hence, the potential incorporation of indirect interactions within functional connections enables the examination of effective interaction between neurons, even with unrecorded units. Third, despite the demonstrated efficacy of various methods to correct for latent variables in the construction of causal graphs [1–4], the applicability of these models in real-world analyses, particularly in the absence of ground truth connectivity, remains uncertain. It is impera-*

tive to underscore that, our study primarily focuses on functional connectivity, which reveals **correlations** instead of causality between neural activities under different stimulus conditions [5]. For this reason, without presuming direct mechanistic interactions, jitter-corrected CCG-based functional connectivity that has been proven effective in capturing the direction of information flow [6, 7] is not significantly impacted by latent variables.’

Furthermore, to avoid potential misinterpretation, we rephrased “excitatory/inhibitory” in the manuscript to “positive/negative”, and subsequently, eFFLb (excitatory feedforward loop) to pFFLb (positive feedforward loop). Nonetheless, in response to your feedback, we have performed a new analysis that illustrates that the presence of unrecorded neurons does not compromise the validity of our main findings on motifs.

To explore the potential influence of joint modulation from these unseen neurons, we conducted additional experiments in which we randomly removed 25% and 50% of neurons from our functional networks. This process effectively created hidden neurons with genuine connections to the observed subnetworks. Our results, presented in the updated Figure S10A, B, reveal that the motif profiles remain fundamentally unchanged when compared to the full networks, with the primary distinction being in the significance level – as is to be expected from sub-sampling the neurons in the dataset. Notably, the pFFLb (formerly called eFFLb) motifs remain the most prominent motif types. These findings affirm that joint modulation from unseen neurons does not introduce any significant impacts that would compromise the integrity and validity of our study’s conclusions, particularly with respect to pFFLb motifs.

Additionally, we conducted further experiments by completely removing all neurons from a higher area (AM), and these results, as displayed in the updated Figure S10C, consistently support our earlier findings. We believe that this supplementary analysis reinforces the robustness and reliability of our conclusions on motifs.

We have added the description for joint modulation analysis in lines 239-240 of the main text: ‘...on various significance levels (Fig.S9) and with potential joint-modulation from unobserved neurons mitigated (Fig.S10)’.

2. Section 2.4 moves from small-scale motifs to modules. The manuscript would benefit from a better integration with the previous part and from a more direct phrasing of its relevance. I believe that the statements such as the one made in line 294 on lower coverage scores of natural images and movies would require some sort of hypothesis on what we would expect from different considerations. The finding presented at this point seems to be well-aligned with coverage simply correlating with pixel correlations, or correlations between low-level features.

In response to this feedback, we have made revisions to the introductory sentence of section 2.4 (lines 273-276): ‘The adoption of network analysis in the investigation of neural mechanisms underlying visual processing offers more than just the exploration of elementary information processing components (e.g., three-neuron motifs). It opens the door to a comprehensive examination of how extensive neural populations, including modular structure, engage in interactions.’

In our revised manuscript, we have modified the paragraph from lines 313-315 and included some additional context and intuition regarding what one might expect to observe in natural images and movies in comparison with artificial stimuli: ‘These three measures all show variation in the module organization for different stimuli (Fig. 4C). One would usually assume that long-range and inter-areal connections are needed for more demanding tasks [17, 18]. However, our results show that for natural images and movies, the modules have a higher propensity to cover only a subset of a brain region than for drifting and static grating stimuli. This is reflected by lower coverage scores and higher purity scores for networks driven by natural images and movies than those driven by drifting and static gratings (Fig. 4C; $p < 10^{-310}$, $p < 10^{-310}$, rank-sum test). Therefore, it appears that enhanced within-areal connections play a beneficial role in visual computation which could be more important in the processing of natural stimuli.’

3. The paper presents the dependence of the participation of specific neurons on stimulus statistics. This statement is most relevant if there is a quantification of the reliability of the estimates of the motifs identified by the authors. The paper presents a fitting to two halves of the stimulus but I am unsure if the two halves contain identical stimuli and if there is a triplet-by-triplet analysis of the stability of the analysis. Ideally, the control should be done to multiple responses to the same stimulus in order to control for stimulus-induced variances,

updated Figure S10: Motif intensity significance sequences of all signed motifs for subnetworks that are generated by randomly removing (A) 25% of neurons, (B) 50% of neurons, (C) all neurons from AM.

but I assume that this is prevented by the design of the experimental session (i.e. scheduling of stimuli).

We acknowledge that our initial submission lacked a thorough explanation of the computational settings. To address this, we want to clarify that when we divided the trials into halves, our intention was to ensure that each half contained identical stimuli. This precaution was taken to prevent any potential bias resulting from variations in stimulus statistics, as mentioned in this comment. Given that we utilized natural scenes in Figure S8, it is important to emphasize that each image was deliberately structured to have an equal number of trials in each half. We have also revised the caption of updated Figure S8B to provide a more comprehensive description of our computational design: ‘(B) Motif sequences of the networks during two halves of the trials of natural scenes. Each row corresponds to a realization of the random split. For each random split, two halves contain identical repeats of each image to control for stimulus-induced variances.’

Related to this, in line 117 the authors report that ‘natural stimuli tend to evoke fewer functional connections than grating stimuli’. Since the analysis assesses total correlations

and it seems that full field gratings are characterized by stronger correlations at the pixel level, it is unclear how much the reduction in joint activations is a consequence of smaller pixel correlations.

As you noted, the CCGs indeed capture total correlations, and the original idea of this work was to quantitatively describe how visual stimuli of varying image statistics induce different functional networks. A point of particular interest resides in the pursuit of a mapping function that relates stimulus statistics to network properties. Although this is beyond the scope of this work, as a future direction, we plan to work on building an image-computable model to predict and explain how the functional networks depend on the visual stimulus.

updated Figure S8: Functional networks constructed from different trials. (A) Motif sequence of the network during all trials of natural scenes. (B) Motif sequences of the networks during two halves of the trials of natural scenes. Each row corresponds to a realization of the random split. For each random split, two halves contain identical repeats of each image to control for stimulus-induced variances. (C) Overlap percentage for pFFLb motifs. For each network, overlap percentage is defined as the percentage of motifs that are found on it and at least one other network. Overlap percentage for each motif type is the average over all networks. Overlap percentage is higher for networks evoked by different trials from the same stimulus than different stimuli; $*p < 0.05$, $**p < 0.001$, rank-sum test.

4. The paper discusses the stimulus dependence of the joint activity patterns. While at some points of the paper it is clearly phrased that it is the dependence on stimulus *statistics* (*type* of stimulus) but ever so often the text makes claims about stimulus dependence of correlations. From the figures it seems that it is always the stimulus type dependence that is investigated but a through clarification of this aspect would be welcome given the importance of this concept. There are studies that attempt to assess how correlations beyond those coming from the similarity/dissimilarity of RF structure dependent on stimulus (characterized as noise correlations) (e.g. apart from the cited Banyai et al paper Lin et al (2015) Neuron; Franke et al (2016) Neuron) and therefore the careful phrasing of the claims would be necessary.

We appreciate the reviewer’s insightful feedback and acknowledge the need for improved clarity regarding the nature of stimulus dependence in our paper. We agree that the term “stimulus dependence” can be ambiguous and has the potential to be misinterpreted. To address this concern, we have revised the entire manuscript to accurately describe the concept

as “stimulus-type dependence” instead, emphasizing that we are specifically referring to the dependence on the type or statistics of the stimulus. We have ensured that our text now consistently conveys this meaning. We have also added the mentioned references (Lin et al (2015) Neuron; Franke et al (2016) Neuron) in line 50. For the same reason, we have updated our title to “Stimulus type shapes the topology of cellular functional networks in mouse visual cortex”.

updated Figure S1: Influence of running speed on functional networks. (A) Distribution of mean speed for each mouse during all visual stimulus types. Error bars show 95% confidence interval across the mean speed of all trials. Each color corresponds to a single mouse. (B) Number of stationary and running trials against the threshold on speed. Data from session 2 during drifting gratings are selected for subsequent analyses due to the comparable number of trials for stationary and running periods. The optimal threshold (1 cm/s) is selected to differentiate stationary and running phases based on the maximal change induced in their number of trials, consistent with the threshold used in previous work [19]. (C) Basic network properties of functional networks of all, stationary and running trials for session 2 during drifting gratings. All metrics are normalized by the maximum for better comparison. (D) Motif significance sequences for networks of all, stationary and running trials for session 2 during drifting gratings as a representative example. (E) WA coverage, WA purity and ARI of stationary and running trials for different stimulus types during session 2.

5. Cortical activity in mice notoriously depends on posture, and movements. Controlling for locomotion/movements seems to be especially crucial during the investigation of joint

activity patterns: locomotion as a latent variable is prone to co-modulate the activities of multiple neurons. Although the authors point out that an analysis was performed to control for movements (pointing out that the analysis is not shown), the details of such an analysis are pertinent to understand the validity of the results.

We agree with you regarding the need to include details on controlling locomotion effects in our study. To address this concern, we would like to emphasize that, indeed, we had conducted a control analysis on running speed, as mentioned in our initial submission. We omitted these results in our initial submission as we initially deemed them to be of lower priority, not contributing substantially to our primary findings. However, in response to your feedback and to the importance of this question, we have incorporated the control analysis for running speed in the updated Figure S1.

To clarify the approach taken, we partitioned the trials into two groups – stationary trials and running trials – and subsequently constructed functional networks separately from trials of each category. While our analysis revealed significant disparities in basic network properties between stationary and running networks, such as density and clustering coefficient (updated Figure S1C), we found that motif profiles and results of module analysis remained consistent regardless of running speed, as demonstrated in the updated Figure S1D, E. We have revised lines 114-115: *‘Our control analysis on running speed (Figure S1) showed that our subsequent observations are indeed determined by the stimulus and not by locomotion.’*

These additional results provide clarity on the relationship between running speed and motifs in our study, supporting the conclusion that our primary findings are not substantially affected by locomotion-related variations.

Minor: line 121: The Vinje & Gallant paper seems to investigate the difference between CRF and CRF + NCRF stimulation rather than natural vs non-natural stimuli, as the text suggests

We thank the reviewer for pointing out this inaccurate reference, we have removed the corresponding citation.

line 123: please define network density more explicitly

We added the following sentence to illustrate the definition of network density, see lines 130-132: *‘While number of nodes (neurons) is predetermined by experimental recording, network density, defined as the present number of connections normalized by the maximum possible number of connections, alone displays the most fundamental properties of a functional network.’*

line 143-146: the paper reports higher signal correlation for neurons within the same motif: I am not sure if this is simply a direct consequence of motifs being defined through cross-correlations between pairs of neurons.

To clarify, lines 154-157 in the revised manuscript compared the signal correlations for connected against disconnected neuron pairs, and furthermore for neuron pairs with excitatory and inhibitory connections. The motif is not introduced here yet, and the comparison is only for pairwise functional connections defined through CCG. Consistent with previous work [20], we find that neuron pairs with similar tuning similarity tend to be (positively) connected.

At the same time, we appreciate the reviewer’s concern about the unfair comparison between neurons within a motif against neurons without any connection regarding results in lines 261-263 and Figure 3C. For this reason, Figure 3 of the original submission compared the signal correlation between neurons within pFFLb motifs against neurons within other significant motifs. We believe that this comparison helps to identify how the relationships between neurons within pFFLb motifs differ from those of other connected neurons (within other significant motif types). By comparing connected neurons within different motif types, we avoid a situation in which we compare signal correlations for connected vs unconnected cell pairs and for within-motif vs non-within-motif. Based on your feedback, we have revised the manuscript on lines 260-261 to make this point more clearly: *‘To further investigate this question, we analyzed the tuning similarity of these motifs. As a control, we compared the tuning profiles of neurons within other significant motifs.’*

line 158: this is an example where the term stimulus dependency seems to be confusing. Unless it is indeed stimulus dependency and not stimulus type dependency.

We appreciate the reviewer’s feedback on the clarity of the manuscript. We have revised the manuscript and made sure that the relevant statements are accurate.

line 169: the reader would benefit from a brief definition of the types of surrogate networks. This is well-motivated technical contribution and it would be great to have an instant grasp on what the different surrogates do

We are grateful for the reviewer’s thoughtful suggestion. We have added a sketch for the surrogate models in the updated Figure 2A. We believe that it provides a better intuition of all the surrogate models adopted in this work.

line 209: some support for the interpretation of degree distribution and neuron pair distribution would be beneficial for the reader.

We believe the updated Figure 2A including the caption makes it easier for the reader to understand these network properties.

line 223 & 231 & 233: the text should be more specific if it is referring to changes in stimulus identity or changes in stimulus statistics

We have reworded the corresponding texts for accuracy, see line 243: ‘... all stimuli (Fig. 3B), those motifs were composed of different neurons for different stimulus types (Fig. 3D,E).’, line 251: ‘... during the same stimulus type than for different types, indicating that the identities of the neurons within the over-represented motifs change as the stimulus type changes.’ and line 252: ‘... as the stimulus type changes...’.

Reviewer 2 (Remarks to the Author): Tang et al investigated the functional connectivity among neurons across visual areas during the absence and presentation of visual stimuli. They designed a novel optimization technique for a modularity score with signed weights. This work provides an interesting set of new analyses on an existing dataset that open up additional questions that upon answering I think would greatly improve the scientific impact of the manuscript (listed below). I have only one major concern with the manuscript: the over-interpretation of the difference between functional connectivity for gratings versus natural stimuli – there are substantial differences in the spatial correlations of the two stimulus sets that could explain a large amount of the differences (see point 4).

We sincerely value your positive feedback and constructive comments on our manuscript. We are delighted that you appreciate the analytical tools applied in our manuscript. We acknowledge your major concern regarding the potential over-interpretation of the difference in functional connectivity between gratings and natural stimuli, specifically due to the influence of spatial correlations. In response, we have thoroughly addressed this concern in the revised manuscript, providing a detailed discussion on the impact of spatial correlations on our findings. We believe this addition strengthens the interpretation and contextualizes our results appropriately.

1) Can the authors briefly describe the running speed control? Do resting-state functionally-connected neurons have similar running speed correlations?

In the revised manuscript, we have provided a concise description of the running speed control analysis (updated Figure S1). We first segregated the trials into two distinct groups: stationary trials and running trials based on the optimal threshold on running speed (1 cm/s). Subsequently, we constructed separate functional networks for each of these categories. Although our analysis unveiled notable differences in fundamental network properties between the stationary and running networks, including variations in density and clustering coefficients, it is worth noting that our key findings regarding motif profiles and modular analysis demonstrated stability across various running speeds, as evidenced in the updated Figure S1D, E. We have revised lines 114-115: ‘Our control analysis on running speed (Figure S1) showed that our subsequent observations are indeed determined by the stimulus and not by locomotion.’

2) Relatedly, how distinct are the functional connections for different stimulus groups? If two neurons are functionally connected based on one stimulus set/resting state, are they more likely to be found to be functionally connected during another stimulus set/resting state?

To address the inquiry raised in the reviewer’s comment, namely, whether different stimulus sets share common functional connections, we employed a metric based on the overlap coefficient. This metric measures the degree to which connections are shared between stimulus types in comparison to the expectation of two random networks with identical densities.

Our analysis reveals that functional connections are notably more similar for related stimulus types, such as natural scenes and movies, as exemplified in the updated Figure S2D.

updated Figure 2: Highly preserved local structure during different types of visual stimuli. (A) Illustration of the surrogate models: Erdős-Rényi model, Degree-preserving model, Pair-preserving model and Signed-pair-preserving model that exhibit a progressive increase in the number of preserved network properties. The preserved properties are the number of nodes (V)/edges (E), degree distribution, neuron-pair distribution and signed-pair distribution, respectively. The real network used to generate the synthetic networks is sampled from an example session during drifting gratings. (B) Relative count of signed neuron pairs with respect to Erdős-Rényi model. All three types of bidirectional pairs are considerably over-represented. (C) 13 types of motifs without edge signs. Examples of signed motifs based on Feedforward Loop structure (motif ID = 6) are shown. Positive-FFL-based (pFFLb) motifs are defined as motifs with at least one FFL structure consisting of all positive connections. Note that pFFLb motifs can be classified into four types based on the number of mutual connections: zero (ID = 6), one (ID = 9, 10, 11), two (ID = 12) and three (ID = 13). In what follows, we use the prefix “p” or “n” to denote signed motif types with only positive connections or only negative connections, respectively. (D) Motif intensity significance sequences of all signed motifs (signed three-neuron subnetworks). Each row shows results for a certain stimulus type and each color corresponds to a certain unsigned motif structure. Motif intensity significance for each signed motif is obtained through its intensity Z score of the empirical network with Signed-pair-preserving model as the reference. The overall sequences are remarkably similar across visual stimulus classes, with six types of over-represented motifs (pFFLb motifs) and three under-represented motifs preserved. It is worth noting that the above-mentioned nine significant motifs only contain excitatory connections.

These findings align with our earlier results on functional motifs and modular structure. Notably, resting-state networks exhibit substantial similarity to networks observed during natural stimuli. Furthermore, networks observed during the presentation of static gratings bear a closer resemblance to those observed during natural stimuli, compared to drifting gratings. These patterns of similarity are consistent with the network density in Figure 1E and motif purity in Figure 4C.

updated Figure S2: Basic properties of functional networks during all visual stimuli. (A) CCG adjacency matrices of a mouse given distinct stimuli (on the same scale as in Fig 1D). (B) Directed degree distributions of a mouse given all stimuli. (C) Across-area *vs* within-area comparisons for the lag τ , duration D and their sum $\tau + D$. (D) Relative overlap coefficient for connections during different visual stimulus groups. To remove density dependence, we used relative overlap coefficient to measure the similarity between connections, defined as overlap coefficient divided by expectation of two random networks with identical densities.

The authors state "While the three-neuron functional connectivity motifs that were most over- or under-represented were the same for all stimuli (Fig. 3B), those motifs were composed of different neurons for different stimuli (Fig. 3D,E)." However, Fig 3D does not show this, and 3E is not understandable to me as someone outside network theory (and is only for the e6 motifs?).

We appreciate the reviewer's feedback regarding the clarity of our manuscript. We understand that our original wording may have led to confusion and acknowledge that there could be inaccuracy in the original caption. In Figure 3D (right), we illustrate the proportion of motifs consisting of three V1 neurons, and this proportion exhibits variations across different stimuli, highlighting the dynamic composition of these motifs. In Figure 3E, we provide an upset plot that illustrates the intersections between more than three sets of motifs. The horizontal bars on the left indicate the number of motifs observed for each stimulus type. Meanwhile, the vertical bars at the top represent the intersection size, which corresponds to the number of motifs observed in the respective stimulus sets indicated at the bottom. Figure 3E effectively demonstrates that the majority of motifs are exclusive to one stimulus type, with the top five vertical bars representing motifs unique to five individual stimulus types. We have modified updated Figure 3D, E to provide a clearer explanation of the relationship between motifs, neurons, and different stimuli in our study: *'Figure 3 (D) ...The variation in regional composition indicates the change of component neurons for pFFLb motifs across different visual stimuli. (E) ...Horizontal bar plot shows the number of signed motif ID = p6 for each type of stimulus while vertical bar plot displays the size of each intersection set. A unique motif is defined as a certain signed motif with three specific neurons, and intersections with less than 20 elements are removed for brevity (see Supplementary Fig. 11 for the*

complete results on all pFFLb motifs).’.

updated Figure 3D,E: (D) Fraction of motifs with at least one V1 neuron or all three V1 neurons during all visual stimuli for six over-represented positive-feedforward-loop-based (pFFLb) motifs. Six colors represent six types of pFFLb motifs, consistent with Fig. 2C. From left to right, motif ID = p6, p9, p10, p11, p12, p13. The variation in regional composition indicates the change of neuron identities for pFFLb motifs across different visual stimuli. (E) Intersections of unique motif sets for motif ID = p6 during six types of stimuli. Horizontal bar plot shows the number of signed motif ID = p6 for each type of stimulus while vertical bar plot displays the size of each intersection set. A unique motif is defined as a certain signed motif with three specific neurons, and intersections with less than 20 elements are removed for brevity (see Supplementary Fig. 11 for the complete results on all pFFLb motifs). A large number of unique motifs appear only during one type of stimuli, demonstrating that even though functional motifs are preserved across visual stimuli, component neurons are changing.

3) Based on the size of the window for the CCG, is it surprising that motif 6 is more common than e.g. 1 or 2? I could imagine that in motif 1 with a large enough window, a connection between the receiving nodes would be identified. It would be useful to show as a function of the window size that the number of FFLs detected does not decrease more than for example the number of 1 and 2 motifs.

It is indeed possible that some motifs identified as motif ID=1,2 or 3 will be classified as motif ID=6 with a larger window size of CCG, because the first three types can be seen as unclosed counterparts of motif ID=6. In the reference Figure 1 we tested whether the motif distribution depends on the window size of CCG. We found that number of connections reflected by network density increases with window size (panel A), and the motif intensity (similar to number of motifs), defined as the summation of the geometric mean of all connections in the motif, also has similar trends (panel B). Therefore, number of motifs generally increases with window size of CCG. However, it is important to note that the number of motifs in surrogate networks also increases in a similar manner. As a result, the significance of motif intensity (real network compared to surrogate networks) does not show a strong dependence on the window size of CCG. This discrepancy again highlights the importance of using surrogate models for analyzing motif and modular structure as seemingly prominent substructures could actually appear at chance level or even below. We have clarified our writing regarding the significance represented in Figure 2C, where we depict motif intensity significance. This measure conveys the significance level for each motif in comparison to the surrogate model.

Also, Figure 3F signed-pair preserving is the reference for Figure 2C – it would be nice if it was introduced in that figure for the reader.

We have added a sketch for the surrogate models in the updated Figure 2A. We believe that it provides a better intuition of all the surrogate models adopted in this work.

4) The across-area analysis uncovers the potential problem in interpreting the functional correlations in the drifting gratings as significant. The drifting gratings are full-field with the same structure at every position. This may produce more functional correlations that are only related to the structure of the stimulus itself. I don't see how interpreting these functional correlations as different from natural scenes is meaningful beyond this fact of stimulus spatial correlation difference, but perhaps the authors could provide some insight

reference Figure 1: Impact of window size of CCG. Session 2 recordings during drifting grating presentations are illustrated as representative examples in this figure. (A) Network density against the window size of CCG (τ_{max}). (B) Motif intensity for motif ID=1, 2, 3 and 6 against window size of CCG (τ_{max}). Note that motif ID=1, 2, and 3 can be considered as unclosed counterparts of motif ID=6. Error region represents the intensity distribution of the corresponding motifs in surrogate networks generated using Signed-pair-preserving model. (C) Motif intensity significance sequences with different window sizes of CCG.

here (Figure 5 continues this point).

We thank the reviewer for this insightful comment. Artificial stimuli such as drifting and static gratings could display distinct image statistics from natural images and movies, and the differences may originate from pixel correlations and other correlations at larger spatial scales. As a result, there could be variations in the number of functional connections. Our rigorous approach to maintaining a uniform number of connections in surrogate networks is a critical aspect of our methodology which ensures that any observed variations in network density do not compromise the robustness of our main results, particularly those related to motifs and modules. For this reason, while the spatial distribution of the stimulus might impact the number of connections detected via our method, these variations do not undermine the core findings of our study.

On the other hand, in this work, we focus on exhibiting how functional networks vary with visual stimuli of different degrees of complexity which stem from properties such as spatial correlations. In other words, in addition to testing **whether** functional networks evoked by gratings and natural stimuli are different, we are essentially exploring **how** these networks differ on multiple network scales. Our major findings support the conclusion that for relatively large neural populations (modules) the connectivity depends on the stimulus complexity while it remains consistent across stimulus types for three-neuron patterns (motif). Despite fluctuations in the number of connections, we find that the motif significance remains consistent across stimuli, revealing an interesting pattern. This phenomenon suggests the presence of shared processing components among stimuli characterized by varying spatial distributions. As a future direction, we plan to build a mapping function from the stimulus statistics to the functional network topology, which ideally is an image-computable model that is beyond the scope of this work.

This statement to me “This emphasizes that a functional module is not strictly the

same as an anatomical brain region: the relationship between these concepts depends on the stimulus-defined context” in light of the differences across stimuli seems unavoidable.

As for the seemingly evident statement “This emphasizes that a functional module is not strictly the same as an anatomical brain region: the relationship between these concepts depends on the stimulus-defined context”, we argue that it is not trivial. Prior to our extensive analysis of functional networks across different stimuli, we lacked insight into the variability of functional networks in response to visual inputs. In particular, the consistent over-representation of pFFLb motifs is unexpected. As for the modular structure, considering extensive work on the functions of visual cortex, one would usually assume that functional modules should highly resemble the anatomical organization during the majority, if not all, of stimuli, with neurons working cohesively inside each visual region. Although higher coverage and lower purity of grating stimuli could be interpreted as the result of their unique spatial distribution, this hypothesis alone cannot explain the contradictory trends in resting-state and flash networks. Nonetheless, our findings indicate that the degree of correspondence between the visual area and functional module is dependent on the specific stimulus type, offering valuable insights into understanding the processing of various visual stimuli.

Also the statement in the discussion about this seems to be an over-interpretation: “On the global scale, however, more complex visual stimuli tend to drive networks into more modular structures with stronger segregation and stronger agreement between structural and functional parcellation. This suggests that functional modules with more spatially segregated structure could be required in more demanding cognitive tasks.”

We acknowledge that there could be over-interpretation in the referred statement. We have modified it in lines 380-384 of the revised manuscript: *‘On the global scale, however, natural stimuli tend to drive networks into more modular structures with stronger segregation and stronger agreement between structural and functional parcellation. Although the extensive inter-areal connections and modules evoked by gratings could result from the intrinsic spatial distribution of the grating stimuli, the functional modules with more spatially segregated structures evoked by natural stimuli could play an important role in the processing of the visual input.’*

5) If the authors do want to claim that different stimuli will always create different functional connections, then perhaps a good analysis would be to split the natural scene or natural movie responses in half and compute the functional connections in the two halves. If these are different, then indeed the functional connections are highly dependent on the stimulus. This is likely true if the neural activity is high-dimensional and/or the number of stimuli presented is low.

We appreciate the reviewer’s attention to this aspect of our study. It is worth noting that we had already undertaken the suggested steps as part of our methodology. In the original manuscript, specifically in Figure S8, we divided the trials during natural scenes into halves, with the explicit aim of ensuring that each half contained identical stimuli. This careful approach was taken to mitigate any potential bias caused by variations in stimulus statistics. Notably, when dealing with natural scenes in Figure S8, we meticulously structured the stimuli to maintain an equal number of trials for each image in each half. The results presented in Figure S8C reveal that networks constructed from different halves of the same stimulus exhibit considerably greater similarity in terms of functional connections and motifs than networks evoked by different stimulus types. Furthermore, we have updated the caption of updated Figure S8B to provide a more detailed and informative description of our computational methodology: *‘Figure S8 (B) ...For each random split, two halves contain identical repeats of each image to control for stimulus-induced variances.’*

6) For the across-area connections in the natural scenes/movies, are they more likely for neurons with similar spatial receptive fields? Or is there something else underlying the across-area functional connectivity?

To answer this question, we employed two distinct metrics to quantify the similarity of receptive fields (RFs): the Euclidean distance between RF centers and the Pearson correlation between RFs (see reference Figure 2).

While it is evident that the distinction between connected and disconnected neuron pairs is more pronounced for within-area connections, it is noteworthy that across-area connections during natural stimuli also exhibit significantly more similar receptive fields than disconnected neuron pairs. This observation underscores the presence of a substantial level of

RF similarity in across-area connections.

We would like to highlight that this increased RF similarity in the context of across-area connections appears to be driven more by the overall resemblance between RFs, rather than being solely attributed to the proximity of the most focused pixels, as indicated by the distance between RF centers.

reference Figure 2: Receptive field comparison for connected and disconnected neuron pairs during natural scenes and movies. Distance measures the Euclidean distance between receptive field centers of the neuron pairs while correlation shows the Pearson correlation of two receptive fields. $** p < 10^{-2}$, $**** p < 10^{-4}$, Student's t-test.

Minor:

What are the definitions of network density and clustering coefficient in Figure 1F? I could not find them in the methods, sorry if I missed it.

We added the following sentence to illustrate the definition of network density, see lines 130-132: *'While number of nodes (neurons) is predetermined by experimental recording, network density, defined as the present number of connections normalized by the maximum possible number, alone displays the most fundamental properties of a functional network.'* As for clustering coefficient, it is introduced in lines 141-142: *'the tendency for triplets of neurons to form closed triangles (e.g., three-neuron motifs 6,7,9-13 in Fig. 2B). This tendency is quantified by the clustering coefficient'*.

Reviewer 3 (Remarks to the Author): In this manuscript, the authors present differences in functional connectivity for different stimulus modalities in the inter-areal network of neurons. The work is based on open-source datasets from the Allen Brain Observatory and comprises simultaneous recordings from an average of several hundred neurons in 6 different areas of the mouse visual system while the mice passively viewed 6 different stimulus modalities. Functional connectivity was estimated based on surrogate corrected sharp peaks and intervals of cross-correlograms (CCGs). A variety of topological analyses were performed on the resulting networks mainly focusing on stimulus modality dependent differences of the 3-neuron connectivity motifs and the community structure. Thereby, the authors showed that despite of similar distribution of motifs across stimulus modalities, the neurons participating in within those motifs differ significantly between modalities. Furthermore, the degree to which the neurons of the networks could be attributed to modules with strong connectivity within and low connectivity between each other also differed significantly between modalities. The results of this study are interesting and build on recent advances in the fields of simultaneous recordings, network connectivity of neurons and network neuroscience.

Nevertheless, some aspects remain unclear, and some analytical issues could significantly affect the results described. Therefore, I suggest a revision of the following points:

Major comments:

1.) In this study CCGs are used to estimate functional connectivity between neurons for different stimulus modalities. However, it has been known for a long time that the distinctness of peaks and troughs in CCGs strongly depends on the number of spikes/ the firing rate of the pair of neurons. In particular, the lower firing rate strongly determines the strength of the correlation, which can neither be corrected by the geometric mean normalization nor the jitter surrogate subtraction. Furthermore, it is evident from Figure 1 that there are significant differences in the average firing rate between stimulus modalities in this dataset. Thus, firing rate differences are a confounding factor for the identified differences between stimulus modalities. It would be highly beneficial to the manuscript to add analyses of rate differences between stimulus modalities and to show that the results described cannot be explained by mere differences in firing rate.

We appreciate the reviewer's concern regarding the use of cross-correlograms (CCGs) in estimating functional connectivity for different stimulus modalities. It is crucial to note that the distinctness of peaks and troughs in **raw** CCGs is indeed influenced by the number of spikes and the firing rate of neuron pairs. We would like to draw your attention to a pertinent study, specifically outlined in the appendix of Adam Kohn and Matthew A. Smith. (2005) [21], which highlights the efficacy of geometric mean normalization in addressing the firing-rate dependency observed in CCG peak height. The choice of an exponent of 0.5 in the normalization process is grounded in the observed slope of 0.71 between raw CCG peak height and the product of firing rates. While we acknowledge the differences in average firing rates among stimulus modalities in our dataset, we contend that the geometric mean normalization, coupled with jitter surrogate subtraction, provides a robust correction for firing rate in connection strengths.

2.) Following the previous point, rate differences affect not only the detection of peaks (and troughs) but even more strongly bias statistical detection of significant connections. This could explain a substantial part of the topological differences between modalities described here. Therefore, additional controls would be very helpful to exclude this possibility such as stratification for number of spikes, a repetition of the analyses with only neurons with very high firing rates or a fixed threshold instead of statistics for connection detection.

It is plausible that variations in firing rates could influence the number of functional connections identified through CCG, potentially impacting our primary findings and conclusions. To thoroughly address this concern, we conducted a comprehensive analysis examining the interplay between functional connections and firing rates across diverse visual stimuli. However, it is important to note that we have opted not to downsample spikes to equalize the firing rate for different stimuli, which seemingly serves as a solid control for firing rate. This choice is based on the understanding that spikes represent invaluable data, and a higher volume of spikes increases our confidence in the constructed functional connections. The CCG method is known to be sensitive to spikes, and retaining the raw spike data enhances the robustness and reliability of our findings.

As shown in the updated Figure S3A, we observed notable fluctuations in firing rates across different visual stimuli and visual areas. To delve into the relationship between functional connectivity and firing rates, we systematically assessed the correlation between the number of functional connections and the firing rate for each neuron, taking into account the specific visual stimulus employed. Our analysis uncovered an intriguing dependency between the correlation patterns of within-area and across-area connections and the visual area and stimulus type.

For instance, during exposure to natural stimuli, we found a robust association between firing rates and within-area connections, while this association was not statistically significant during flashes for most areas. In the context of resting-state networks, we even observed a scenario where neurons with higher firing rates exhibited fewer across-area connections. Notably, the discrepancy between within-area and across-area connections appeared to intensify further during resting-state and natural stimuli (updated Figure S3B).

These findings collectively underscore the stimulus-dependent nature of the relationship between firing rates and functional connections, **thereby emphasizing that the disparities in firing rates alone do not offer a comprehensive explanation for the distinct**

A

B

updated Figure S3: Firing rate and its relationship with functional connection. (A) Firing rate distributions for each visual area during all visual stimuli. (B) Correlation coefficient between a neuron’s firing rate and its number of within-area/across-area functional connections. ns $p > 0.05$, $*p < 0.05$, $**p < 0.01$, $***p < 0.001$, $****p < 0.0001$, Wald Test for each correlation value, Modified Asymptotic Test [22] for comparing overlapping correlations (firing rate as the common variable).

network topologies evoked by various visual stimuli. We believe that this nuanced understanding of the intricate interplay between firing rates and functional connections significantly enhances the depth and robustness of our study.

We added the following text to illustrate this point, see lines 134-136: ‘*We did a thorough analysis of the firing-rate dependence of functional connections and found that the difference in the number of connections during various stimulus types cannot be explained by the difference in firing rate (Figure S3).*’

We also added a detailed explanation of the statistical test adopted in the updated Figure S3 in Methods section: ‘*We use the modified asymptotic (MA) test for comparing correlations between the firing rate of a neuron and its number of within-area/across-area connections [22]. Since the two correlations in the test involve a common variable (firing rate), a test for overlapping correlations is adopted. Suppose r_1 , r_2 and r_3 are the correlations between firing rate and within-area connections, firing rate and across-area connections, and within-area and across-area connections, respectively. The MA method defines the confidence limits (L, U) for the correlation difference $r_1 - r_2$ as*

$$L = r_1 - r_2 - \sqrt{(r_1 - l_1)^2 + (u_2 - r_2)^2 - \widehat{\text{corr}}(r_1, r_2)(r_1 - l_1)(u_2 - r_2)} \quad (1)$$

$$U = r_1 - r_2 + \sqrt{(u_1 - r_1)^2 + (r_2 - l_2)^2 - \widehat{\text{corr}}(r_1, r_2)(u_1 - r_1)(r_2 - l_2)} \quad (2)$$

where l, u are the confidence limits for single correlation r with a significance level α

$$l = r - z_{\frac{\alpha}{2}} \sqrt{\widehat{\text{var}}(r)} \quad (3)$$

$$u = r + z_{\frac{\alpha}{2}} \sqrt{\widehat{\text{var}}(r)} \quad (4)$$

where $\widehat{\text{var}}(r) = \frac{(1-r^2)^2}{n}$ and n is the sample size. The correlation between the two correlations $\widehat{\text{corr}}(r_1, r_2)$ can be estimated by

$$\widehat{\text{corr}}(r_1, r_2) = \frac{\widehat{\text{cov}}(r_1, r_2)}{\sqrt{\widehat{\text{var}}(r_1)\widehat{\text{var}}(r_2)}} \quad (5)$$

where covariance $\widehat{\text{cov}}(r_1, r_2)$ between the two correlations can be obtained through

$$\widehat{\text{cov}}(r_1, r_2) = \frac{(r_3 - \frac{1}{2}r_1r_2)(1 - r_1^2 - r_2^2 - r_3^2) + r_3^3}{n} \quad (6)$$

Therefore, if $L(r_1, r_2) > 0$, r_1 is considered significantly higher than r_2 with significance level α ; on the contrary, if $L(r_2, r_1) > 0$, r_2 is considered significantly higher than r_1 with significance level α .

3.) In the manuscript clear differences are shown between grating (static and moving) and other stimulus modalities. Grating stimuli are well known to cause oscillatory couplings between neurons, which also result in sharp peaks and intervals. It would be important to show that the described differences are not explained by differences in oscillatory connections or if they are, to show these differences.

It is indeed well-known that drifting gratings evoke oscillations between neurons. In our previous work we have investigated how different conditions of drifting gratings shape the functional connectivity of mouse visual cortex [6], and how gamma oscillations correlate to jitter-corrected CCG sharp peaks [23]. To test whether there is an oscillation in neural activity evoked by various stimuli in this experiment, we computed the power spectrum of spike trains for each visual area (see reference Figure 3). Nevertheless, our analysis indicates an absence of statistically significant oscillations in neural activity when exposed to gratings in comparison to alternative stimuli. This incongruity can be attributed to three primary factors. Firstly, there exists a disparity in the animal models utilized, with prior studies [23] concentrating on monkeys, while our investigation is centered around mice. Secondly, the variation in outcomes may be attributed to the methodological approach, specifically the practice of averaging across distinct stimulus conditions, including orientation and spatial frequency, particularly in the context of gratings. Thirdly, oscillatory patterns have been noted to demonstrate increased amplitude during anesthesia and epilepsy in contrast to awake periods [24]. Hence, the primary findings remain unaffected by oscillatory coupling during specific stimuli.

reference Figure 3: Average power spectrum of spike trains for each visual area from session 2 during various types of visual stimuli. Power spectrum is obtained through multi-taper Fast Fourier Transform. Due to the limited presentation duration of other stimuli (250 ms), only spike spectra during resting state, drifting gratings and natural movies are shown.

4.) On page 4 in the lines 168-172 the authors describe the used surrogate networks for normalization and statistical testing of network measures throughout the manuscript. However, in contrast to fMRI-based networks, neurophysiological recordings do not represent the whole network, but are patchy spatial subsamples. Moreover, the number of connections between neurons decreases strongly with distance. Therefore, it is important to resemble the distance dependent connectivity when generating surrogate networks for topological analyses of networks of neurons as described in: (Gerhard et al., 2011; Dann et al., 2016). The strong spatial subsampling could otherwise explain a significant part of the described motif distributions as well as the community structures.

We thank the reviewer for raising a pertinent concern regarding the surrogate networks used for normalization and statistical testing in our manuscript. The distinction between neurophysiological recordings, which represent patchy spatial subsamples, and fMRI-based

networks is indeed a crucial point. We acknowledge the necessity of introducing spatial information into our surrogate model to address the spatial dependencies present in neurophysiological data.

To rectify this issue, we have modified our surrogate network model. In addition to preserving all network properties as in the signed-pair-preserving model, we have now also maintained the anatomical distance distribution of the surrogate networks to mirror that of the empirical network. Recognizing that reproducing an identical distance distribution is infeasible after random shuffling, we have preserved the discretized distance distribution (as illustrated in Figure S4A), ensuring that the generated surrogate networks maintain the same number of connected neuron pairs within each distance bin.

As a result, we have found that the most significant motifs, including pFFLb motifs, remain consistent using the anatomical-distance-preserving model, as indicated in the updated Figure S5A. Furthermore, we have identified another noteworthy three-neuron pattern involving one neuron both sending negative connections to and receiving positive connections from two other neurons, which emerges as an additional significant motif type.

updated Figure S5: Network properties using anatomical-distance-preserving model. Only neurons with recorded anatomical locations are included. (A) Motif sequence of the networks during 6 visual stimuli using anatomical-distance-preserving model. (B) Topological structure of functional connectivity of a mouse during six types of visual stimuli with neurons colored by area. (C) Signal correlation for within-module and across-module connections. $**** p < 10^{-4}$, Student's t-test. (D) WA coverage, WA purity and ARI during six visual stimuli.

The use of the distance-preserving model for calculating motif significance enables us to conclude that the spatial distribution of neurons does not account for the prominence of pFFLb motifs.

Regarding the identification of modular structures, the distance-preserving model also yields consistent outcomes with the signed-pair-preserving model. As detailed in the updated Figure S5B,D, we tend to identify more pure modules during natural stimuli, while static

and drifting gratings result in modules that cover a larger fraction of single visual areas. Our findings further support the conclusion that neurons within the same module exhibit more similar functions than neurons in different modules (as demonstrated in updated Figure S5C), in accordance with previous results.

We added the following text to illustrate this point, see lines 186-191: *‘Furthermore, we have consistently observed and validated our findings by maintaining the discretized distribution of anatomical distances between neurons (as shown in Figure S5). This rigorous approach underscores the robustness of our primary conclusions, affirming that they are an accurate representation of the inherent characteristics of the functional networks within the mouse visual cortex. In light of the limited data on the anatomical location of neurons, we have adopted the Signed-pair-preserving model for all analyses in the subsequent sections of our study.’*

We appreciate the reviewer’s insightful feedback, and we believe these revisions significantly strengthen the rigor and validity of our study.

Minor comments:

1.) In the Introduction in line 34 the authors state that: “neurons within the network adapt quickly to different stimuli”. However, the study cited is a PET study in which the activity of neurons is not directly measured, therefore this formulation is misleading.

We have made the necessary revisions to our citations to ensure improved accuracy and clarity, as follows:

Hermes, Dora, et al. “Stimulus dependence of gamma oscillations in human visual cortex.” *Cerebral cortex* 25.9 (2015): 2951-2959.

Ruff, Douglas A., and Marlene R. Cohen. “Stimulus dependence of correlated variability across cortical areas.” *Journal of Neuroscience* 36.28 (2016): 7546-7556.

2.) In the Introduction in the lines 60-61 the authors state that: “Functional connectivity at this finer scale is less well-studied due to challenges in simultaneous recordings from large populations of neurons with high spatial and temporal resolution.” This is not entirely correct, since a number of studies have been published on this subject, which should be cited and the sentence reworded accordingly: (Fujisawa et al., 2008; Yu et al., 2008; Berger et al., 2010; Gerhard et al., 2011; Dann et al., 2016; Nigam et al., 2016; Humphries, 2017).

We have reworded the statement in lines 60-62 as follows and included the corresponding citations:

‘Studying functional connectivity at this finer scale presents significant challenges due to technical limitations in simultaneously recording from large populations of neurons with high spatial and temporal resolution. Despite these challenges, prior work has shown that functional connectivity...’

Statement in line 62 is reworded: *‘1) shows frequency dependency (Frien et al., 2000; Dann et al., 2016)’*.

We also added a summary of the mentioned work in lines 65-66: *‘...whose activities do not. Other studies looked into assembly neurons (Berger, 2010), network dynamics (Humphries, 2017), small-worldness (Yu et al., 2008; Gerhard et al., 2011) and rich-club structure (Nigam et al., 2016)’*.

3.) Following on from point 2) in lines 65-67 (Dann et al., 2016) should be cited.

We have included the relevant citation in line 62 and reworded lines 67-68: *‘they have not included detailed analyses of networks spanning multiple brain regions evoked by distinct stimulus types’*.

4.) Following on from point 2) in lines 69-71 (Fujisawa et al., 2008) should be cited.

We have reworded lines 69-72: *‘, either focused on ... network structure, or investigated the short-term adaptation of pairwise functional connections (Fujisawa et al., 2008), lacking a comprehensive analysis of the whole network.’*

5.) The order of supplementary figures mentioned in the text is confusing because some later figures are described before earlier figures. I consider reordering the supplementary figures.

We have taken steps to rectify this issue, ensuring that all supplementary figures are now presented in the order of their respective appearances in the main text.

6.) In the Methods, the used jitter correction method sounds very interesting, but is not properly described in the manuscript. It would be very helpful to add a short description of the method.

We added a description of the jitter-correction method in lines 476-479: *‘The spike trains are split into short jitter windows with equal size and the spike-jitter method perturbs the spike timing while maintaining the number of spikes within each jitter window to disrupt fast-time correlations. Jitter correction has been widely used to eliminate the shared trial-to-trial variability on a slower time scale [25]. The jitter-corrected CCG...’*.

At the start of line 480 we introduced the size of jitter window used in the manuscript: *‘The size of the jitter window is defined as 25 ms.’*

7.) In the Methods, it is not entirely clear how CCGs with 0 lag peak (or trough) were dealt with.

We have reworded lines 492-496: *‘CCG peaks with zero time lag are explained by dynamical relaying mechanisms instead of common drive [26, 27], thus are usually treated as bidirectional connections [28]. Considering our CCG method can detect bidirectional connections with non-zero peaks, for connections with zero time lag, we retain the primary direction of communication, defined as the direction with a higher significance level. The alternative direction retains its significance when a non-zero significant peak or interval is observed.’*

8.) In the Methods in the lines 463-465 the authors state that: “For each neuron pair, we only keep trials in which spike trains of both neurons have a normalized entropy of at least 0.9.” Discarding trials sounds like it could lead to a biased results.

We thank the reviewer for the detailed attention to our methods. The usage of normalized entropy was to reduce the bias brought by the false edge detection due to the lack of spike. We have reworded the lines 505-510: *‘It has been shown that a sufficiently large number of spikes for both neurons are necessary to make the detection reliable [29], which is directly presented by the CCG. To eliminate the bias brought by insufficient spikes, we used normalized entropy, defined as the Shannon entropy of the CCG divided by the entropy of a constant sequence with the same length (maximum of entropy), to keep only reliable connections where both neurons have enough spikes. We used a threshold of 0.9 on normalized entropy, and most removed connections are found during flashes and are due to the shortage of trials.’*

9.) In Figure 3D and in Figure 4C panels would be better understandable with x-axis labels.

10.) In Figure 3E the symbols for the different modalities are very hard to recognize.

To reviewer’s minor comments 9) and 10): We have modified Figure 3D and Figure 4C as well as the caption for Figure 3E to enhance their readability, see updated Figure 3 and updated Figure 4.

Berger D, Borgelt C, Louis S, Morrison A, Grün S (2010) Efficient Identification of Assembly Neurons within Massively Parallel Spike Trains. *Comput Intel Neurosc* 2010:439648.

Dann B, Michaels JA, Schaffelhofer S, Scherberger H (2016) Uniting functional network topology and oscillations in the fronto-parietal single unit network of behaving primates. *Elife* 5:e15719 Available at: <http://elifesciences.org/lookup/doi/10.7554/eLife.15719>.

Fujisawa S, Amarasingham A, Harrison MT, Buzsáki G (2008) Behavior-dependent short-term assembly dynamics in the medial prefrontal cortex. *Nat Neurosci* 11:823–833 Available at: <http://www.ncbi.nlm.nih.gov/pmc/articles/PMC2562676/>.

Gerhard F, Pipa G, Lima B, Neuenschwander S, Gerstner W (2011) Extraction of Network Topology From Multi-Electrode Recordings: Is there a Small-World Effect? *Front Comput Neurosci* 5:4 Available at:

http://www.frontiersin.org/Computational_Neuroscience/10.3389/fncom.2011.00004/abstract.

Humphries MD (2017) Dynamical networks: Finding, measuring, and tracking neural population activity using network science. *Netw Neurosci* 1:324–338

Available at: http://www.mitpressjournals.org/doi/abs/10.1162/NETN_a.00020.

Nigam S, Shimono M, Ito S, Yeh F-C, Timme N, Myroshnychenko M, Lapish CC, Tosi Z, Hottowy P, Smith WC, Masmanidis SC, Litke AM, Sporns O, Beggs JM (2016) Rich-Club Organization in Effective Connectivity among Cortical Neurons. *The J Neurosci* 36:670–684 Available at: <http://www.jneurosci.org/cgi/doi/10.1523/JNEUROSCI.2177-15.2016>.

Yu S, Huang D, Singer W, Nikolic D (2008) A Small World of Neuronal Synchrony. *Cerebral Cortex* 18:2891–2901

Available at: <http://www.cercor.oxfordjournals.org/cgi/doi/10.1093/cercor/bhn047>.

All suggested references are added and properly cited in the revised manuscript.

updated Figure 3: Same motifs and similar patterns are organized from different neurons. (A) Average absolute motif significance (absolute Z score of intensity) across all signed motifs against network density. Similar to within-area fraction and clustering coefficient (Fig. 1F), there is also a logarithmic relationship between motif significance and density. (B) Pair-wise correlation of normalized motif intensity distribution for six visual stimuli. Extremely high correlation proves the similar motif presence during different types of stimuli. (C) Signal correlation during four visual stimuli (except for resting state) for within-pFFLb-motif, within-other-motif connections and others. Other over-represented motifs are determined using a significance level of 99%. $*p < 0.05$, $****p < 0.0001$, rank-sum test. (D) Fraction of motifs with at least one V1 neuron or all three V1 neurons during all visual stimuli for six over-represented positive-feedforward-loop-based (pFFLb) motifs. Six colors represent six types of pFFLb motifs, consistent with Fig. 2C. From left to right, motif ID = p6, p9, p10, p11, p12, p13. The variation in regional composition indicates the change of neuron identities for pFFLb motifs across different visual stimuli. (E) Intersections of unique motif sets for motif ID = p6 during six types of stimuli. Horizontal bar plot shows the number of signed motif ID = p6 for each type of stimulus while vertical bar plot displays the size of each intersection set. A unique motif is defined as a certain signed motif with three specific neurons, and intersections with less than 20 elements are removed for brevity (see Supplementary Fig. 11 for the complete results on all pFFLb motifs). A large number of unique motifs appear only during one type of stimuli, demonstrating that even though functional motifs are preserved across visual stimuli, component neurons are changing. (F) Multiple motif intensity significance sequences were obtained through four different reference models for natural movies as the representative stimulus type: Erdős-Rényi model, Degree-preserving model, Pair-preserving model and Signed-pair-preserving model with an increasing number of preserved network properties (Methods). Colors are consistent with (D) and Fig. 2C, and connectivity pattern is shown only once for each type of most significant signed motif for brevity. Empirical functional networks are progressively more similar to surrogate networks from left to right.

References

- [1] Mélanie Boly, Rosalyn Moran, Michael Murphy, Pierre Boveroux, Marie-Aurélié Bruno, Quentin Noirhomme, Didier Ledoux, Vincent Bonhomme, Jean-François Brichant, Giulio Tononi, et al. Connectivity changes underlying spectral eeg changes during propofol-induced loss of consciousness. *Journal of Neuroscience*, 32(20):7082–7090, 2012.
- [2] Heba Elsegai, Helen Shiells, Marco Thiel, and Björn Schelter. Network inference in the presence of latent confounders: The role of instantaneous causalities. *Journal of Neuroscience Methods*, 245:91–106, 2015.
- [3] Sindy Löwe, David Madras, Richard Zemel, and Max Welling. Amortized causal discovery: Learning to infer causal graphs from time-series data. In *Conference on Causal Learning and Reasoning*, pages 509–525. PMLR, 2022.
- [4] Xu Wang and Ali Shojaie. Causal discovery in high-dimensional point process networks with hidden nodes. *Entropy*, 23(12):1622, 2021.
- [5] Karl J. Friston. Functional and effective connectivity: A review. *Brain Connectivity*, 1(1):13–36, 2011.
- [6] Joshua H Siegle, Xiaoxuan Jia, Séverine Durand, Sam Gale, Corbett Bennett, Nile Graddis, Gregory Heller, Tamina K Ramirez, Hannah Choi, Jennifer A Luviano, et al. Survey of spiking in the mouse visual system reveals functional hierarchy. *Nature*, 592(7852):86–92, 2021.

updated Figure 4: Distinct modular structures during different types of visual stimuli. (A) Topological structure of functional connectivity of a mouse during six types of visual stimuli with neurons colored by area. The color of each connection shows its sign with red denoting excitatory connection and blue representing inhibitory correlation. The community partition is obtained through modified Modularity for signed networks (see Methods). We computed the Z score of Modularity with Signed-pair-preserving model as the reference to show the degree to which functional network has a modular structure. Networks during gratings and natural stimuli show significant modular structure. (B) We used three measures to reveal the modular structure regarding visual area from different perspectives. Coverage and purity are module-level measures, where the former marks the degree to which the module covers any visual area, while the latter measures the degree to which all neurons in the module are from the same visual area. We computed the average coverage and purity weighted by module size to show the overall properties of the whole functional network (see Methods). Adjusted Rand Index (ARI), a network-level measure, was also used to quantify the difference between module partition and visual areal organization. The weighted average (WA) coverage is 0.375 and 1 (ranges from 0 to 1), WA purity is 0.333 and 1 (ranges from 0 to 1) and ARI is -0.03 and 1 (ranges from -0.5 to 1) for the corresponding two toy examples visualizing the ‘low’ and ‘high’ cases for the measure. (C) WA coverage, WA purity and ARI during six visual stimuli. The error bars show the confidence intervals over all mice obtained with non-parametric bootstrap method. In general, there tend to be fewer and larger modules with higher coverage during grating stimuli, whereas we usually find more and smaller modules with higher purity during natural stimuli. As a result, ARI is lower during resting state and flash while higher during grating and natural stimuli.

- [7] Xiaoxuan Jia, Joshua H Siegle, Séverine Durand, Gregory Heller, Tamina K Ramirez, Christof Koch, and Shawn R Olsen. Multi-regional module-based signal transmission in mouse visual cortex. *Neuron*, 110(9):1585–1598, 2022.
- [8] Catherine Duclos, Danielle Nadin, Yacine Mahdid, Vijay Tarnal, Paul Picton, Giancarlo Vanini, Goodarz Golmirzaie, Ellen Janke, Michael S Avidan, Max B Kelz, et al. Brain network motifs are markers of loss and recovery of consciousness. *Scientific reports*, 11(1):3892, 2021.
- [9] Danielle Nadin, Catherine Duclos, Yacine Mahdid, Alexander Rokos, Mohamed Badawy, Justin Le-tourneau, Caroline Arbour, Gilles Plourde, and Stefanie Blain-Moraes. Brain network motif topography may predict emergence from disorders of consciousness: a case series. *Neuroscience of Consciousness*, 2020(1):niaa017, 2020.
- [10] Joseph B Dechery and Jason N MacLean. Functional triplet motifs underlie accurate predictions of single-trial responses in populations of tuned and untuned v1 neurons. *PLoS computational biology*, 14(5):e1006153, 2018.
- [11] Sarah E Morgan, Sophie Achard, Maite Termenon, Edward T Bullmore, and Petra E Vértes. Low-

- dimensional morphospace of topological motifs in human fmri brain networks. *Network Neuroscience*, 2(02):285–302, 2018.
- [12] Federico Battiston, Vincenzo Nicosia, Mario Chavez, and Vito Latora. Multilayer motif analysis of brain networks. *Chaos: An Interdisciplinary Journal of Nonlinear Science*, 27(4):047404, 2017.
- [13] Ron Milo, Shai Shen-Orr, Shalev Itzkovitz, Nadav Kashtan, Dmitri Chklovskii, and Uri Alon. Network motifs: simple building blocks of complex networks. *Science*, 298(5594):824–827, 2002.
- [14] Nadav Kashtan, Shalev Itzkovitz, Ron Milo, and Uri Alon. Topological generalizations of network motifs. *Physical Review E*, 70(3):031909, 2004.
- [15] Sen Song, Per Jesper Sjöström, Markus Reigl, Sacha Nelson, and Dmitri B Chklovskii. Highly nonrandom features of synaptic connectivity in local cortical circuits. *PLoS Biology*, 3(3):e68, 2005.
- [16] Uri Alon. Network motifs: theory and experimental approaches. *Nature Reviews Genetics*, 8(6):450–461, 2007.
- [17] Marius Schneider, Ana Clara Broggin, Benjamin Dann, Athanasia Tzanou, Cem Uran, Swathi Sheshadri, Hansjörg Scherberger, and Martin Vinck. A mechanism for inter-areal coherence through communication based on connectivity and oscillatory power. *Neuron*, 109(24):4050–4067, 2021.
- [18] Conrado A Bosman, Jan-Mathijs Schoffelen, Nicolas Brunet, Robert Oostenveld, Andre M Bastos, Thilo Womelsdorf, Birthe Rubehn, Thomas Stieglitz, Peter De Weerd, and Pascal Fries. Attentional stimulus selection through selective synchronization between monkey visual areas. *Neuron*, 75(5):875–888, 2012.
- [19] Allen Institute. Allen brain observatory – neuropixels visual coding, technical white paper: overview. https://brainmapportal-live-4cc80a57cd6e400d854-f7fdcae.divio-media.net/filer_public/80/75/8075a100-ca64-429a-b39a-569121b612b2/neuropixels_visual_coding_-_white_paper_v10.pdf, 2019.
- [20] Matthew A Smith and Adam Kohn. Spatial and temporal scales of neuronal correlation in primary visual cortex. *Journal of Neuroscience*, 28(48):12591–12603, 2008.
- [21] Adam Kohn and Matthew A Smith. Stimulus dependence of neuronal correlation in primary visual cortex of the macaque. *Journal of Neuroscience*, 25(14):3661–3673, 2005.
- [22] Guang Yong Zou. Toward using confidence intervals to compare correlations. *Psychological methods*, 12(4):399, 2007.
- [23] Xiaoxuan Jia, Seiji Tanabe, and Adam Kohn. Gamma and the coordination of spiking activity in early visual cortex. *Neuron*, 77(4):762–774, 2013.
- [24] Gyorgy Buzsaki and Andreas Draguhn. Neuronal oscillations in cortical networks. *science*, 304(5679):1926–1929, 2004.
- [25] Asohan Amarasingham, Matthew T Harrison, Nicholas G Hatsopoulos, and Stuart Geman. Conditional modeling and the jitter method of spike resampling. *Journal of Neurophysiology*, 107(2):517–531, 2012.
- [26] Raul Vicente, Leonardo L Gollo, Claudio R Mirasso, Ingo Fischer, and Gordon Pipa. Dynamical relaying can yield zero time lag neuronal synchrony despite long conduction delays. *Proceedings of the National Academy of Sciences*, 105(44):17157–17162, 2008.
- [27] Leonardo L Gollo, Claudio Mirasso, Olaf Sporns, and Michael Breakspear. Mechanisms of zero-lag synchronization in cortical motifs. *PLoS computational biology*, 10(4):e1003548, 2014.
- [28] Benjamin Dann, Jonathan A Michaels, Stefan Schaffelhofer, and Hansjörg Scherberger. Uniting functional network topology and oscillations in the fronto-parietal single unit network of behaving primates. *Elife*, 5:e15719, 2016.
- [29] Ryota Kobayashi, Shuhei Kurita, Anno Kurth, Katsunori Kitano, Kenji Mizuseki, Markus Diesmann, Barry J Richmond, and Shigeru Shinomoto. Reconstructing neuronal circuitry from parallel spike trains. *Nature communications*, 10(1):4468, 2019.

REVIEWERS' COMMENTS

Reviewer #1 (Remarks to the Author):

The authors introduced a number of updates to the manuscripts and addressed many of the points raised in the previous round of review. As a result, the clarity of the manuscript was substantially improved and the new controls now provide more precise insights.

I have one remaining major concern about the conclusions of the paper. Both Reviewer 2 and points #2 & #4 of my review raised the concern that the structural differences between natural and grating stimuli might seriously affect the conclusions of the paper. This aspect of #3 _____ were not directly addressed in the response. However, the authors reflect on the concern by updating the discussion: 'Although the extensive inter-areal connections and modules evoked by gratings could result from the intrinsic spatial distribution of the grating stimuli (...)' This limitation considerably affects the scope of the paper and shall be better reflected in the way the abstract and introduction are phrased. For instance, the text 'functional connectivity can vary as the neurons within the network adapt quickly to different stimuli [12, 13]. This motivated us to ask whether and how different stimuli might engage different functional networks with single-neuron resolution within the visual cortex' seems misleading as it suggests that adaptation to different stimuli will be investigated. However, the presented analysis does not exclude the possibility that the identified motifs simply emerge as a result of joint drive. In the light of the above excerpt from the discussion, the statement 'Our results indicate a surprising degree of stimulus-dependence to the topological structure of functional networks between individual neurons in visual cortex.' seems strong.

Minor:

As the two-element features identified here essentially reflect correlated variability, I believe that a highly relevant precursor of this study was Fiser, Chiu & Weliky (2004) Nature.

Reviewer #2 (Remarks to the Author):

Thank you very much to the authors for thoroughly addressing all my concerns. I would recommend including Reference Figure 2 in Figure 5D/E if possible, I think it is increased evidence for your findings, and helpful for the reader. Thank you also for sharing all of the code for the analyses and the figures. It might be nice to include a minimal notebook for a user to run the motif analysis on their own dataset, I think neuroscientists would generally find it useful.

Reviewer #2 (Remarks on code availability):

The readme is clear for installation. I did not run the code because it required downloading their datasets.

Reviewer #3 (Remarks to the Author):

The authors have thoroughly revised their manuscript and have addressed the concerns I raised in my initial review. By adding additional analyses and controls, the manuscript has substantially improved and provides an important contribution to the understanding of the differences in functional connectivity for different stimulus modalities in the inter-areal network of neurons.

In detail, the added correlation analyses between number of connections and the firing rate per neuron and per stimulus modality provide a significant improvement of the study. The finding that firing rates are in some cases positively correlated with the number of connections, while in other cases they are not or even negatively correlated, will be of great interest to large parts of the field.

Also, the additional statistical tests by means of anatomical distance distribution preserving surrogate networks, which show that the described motif sequence cannot be explained by spatial subsampling, significantly increases the validity of the results and are an important contribution to the field.

Response letter

May 7, 2024

Response to Reviewers' comments:

Reviewer 1 (Remarks to the Author): The authors introduced a number of updates to the manuscripts and addressed many of the points raised in the previous round of review. As a result, the clarity of the manuscript was substantially improved and the new controls now provide more precise insights.

Thank you for acknowledging our efforts to address your previous concerns and improve the manuscript's clarity. We are glad that the updates made have led to clearer insights.

I have one remaining major concern about the conclusions of the paper. Both Reviewer 2 and points #2 & #4 of my review raised the concern that the structural differences between natural and grating stimuli might seriously affect the conclusions of the paper. This aspect of #3 were not directly addressed in the response. However, the authors reflect on the concern by updating the discussion: 'Although the extensive inter-areal connections and modules evoked by gratings could result from the intrinsic spatial distribution of the grating stimuli (...)' This limitation considerably affects the scope of the paper and shall be better reflected in the way the abstract and introduction are phrased. For instance, the text 'functional connectivity can vary as the neurons within the network adapt quickly to different stimuli [12, 13]. This motivated us to ask whether and how different stimuli might engage different functional networks with single-neuron resolution within the visual cortex' seems misleading as it suggests that adaptation to different stimuli will be investigated. However, the presented analysis does not exclude the possibility that the identified motifs simply emerge as a result of joint drive. In the light of the above excerpt from the discussion, the statement 'Our results indicate a surprising degree of stimulus-dependence to the topological structure of functional networks between individual neurons in visual cortex.' seems strong.

We thank the reviewer for raising this important concern. To address this issue, we revised lines 20-21: '*functional connectivity can vary as the neurons within the network adjust their firing patterns quickly to different stimuli [12, 13]. This motivated us to ask whether and how different stimuli might engage different functional networks with single-neuron resolution within the visual cortex*' and lines 27-28: '*Our results indicate that distinct stimulus types can lead to different topological structures of functional networks between individual neurons in visual cortex*'.

Minor: As the two-element features identified here essentially reflect correlated variability, I believe that a highly relevant precursor of this study was Fiser, Chiu & Weliky (2004) Nature.

We have included the relevant citation in lines 35-37: 'These efforts are complicated by the fact that different types of stimuli lead to different dynamical patterns of neural activity and to different degrees of correlation between neurons (Fiser, 2004, etc.)'.

Reviewer 2 (Remarks to the Author): Thank you very much to the authors for thoroughly addressing all my concerns. I would recommend including Reference Figure 2 in Figure 5D/E if possible, I think it is increased evidence for your findings, and helpful for the reader.

We appreciate your suggestion to incorporate Reference Figure 2 into Figure 5D/E to bolster the evidence for our findings and enhance reader comprehension. We have added Reference Figure 2 along with corresponding results on modular structure as Figure 5F in the revised manuscript. The individual level comparison in updated Figure 5F shows both the similarity and dissimilarity between brain area and functional module. Within

and across both areas and modules, functionally connected neurons are more likely to share similar receptive field properties. However, the across-area connections are less dependent on receptive field similarity than within-area connections, while the opposite trend is observed for modular structure. This observation thus renders interesting insights on structures and functions of areas and modules. While different areas are known to have distinct functions and between-area interactions could be attributed to communications, functional modules are finer-scale parcellations and multiple modules could share relatively similar functions. We believe that this supplementary analysis provides more intuition of the functional modules.

We have added the description for updated Figure 5F in lines 332-334 of the main text: *‘Functional interactions tend to be found between neurons with more similar receptive fields across most scenarios concerning areal and modular structures, although how strongly interactions depend on receptive field similarity varies (Fig. 5F). This shows that functional modules are finer-scale partitions, displaying both the similarity and dissimilarity between anatomical and functional parcellation.’*

Thank you also for sharing all of the code for the analyses and the figures. It might be nice to include a minimal notebook for a user to run the motif analysis on their own dataset, I think neuroscientists would generally find it useful.

We acknowledge your appreciation for the provided code and your suggestion to include a minimal notebook for conducting motif analysis on user datasets. We will include such notebook in the Github repository promptly after organizing the code, well before the publication of the manuscript.

Reviewer 3 (Remarks to the Author): The authors have thoroughly revised their manuscript and have addressed the concerns I raised in my initial review. By adding additional analyses and controls, the manuscript has substantially improved and provides an important contribution to the understanding of the differences in functional connectivity for different stimulus modalities in the inter-areal network of neurons. In detail, the added correlation analyses between number of connections and the firing rate per neuron and per stimulus modality provide a significant improvement of the study. The finding that firing rates are in some cases positively correlated with the number of connections, while in other cases they are not or even negatively correlated, will be of great interest to large parts of the field.

We appreciate the reviewer’s acknowledgment of the revisions made to the manuscript and their recognition of the improved quality. We are pleased that the reviewer found the added correlation analyses to be a significant improvement to the study. Although a direct interpretation of the complicated relationships is difficult, we believe that additional investigation into how firing rate and functional interactions co-encode input information for different stimulus types in follow-up work could provide valuable insight into visual processing.

Also, the additional statistical tests by means of anatomical distance distribution preserving surrogate networks, which show that the described motif sequence cannot be explained by spatial subsampling, significantly increases the validity of the results and are an important contribution to the field.

We appreciate the recognition of the value of the additional statistical tests. We agree that they enhance the credibility of our findings and offer valuable insights.

Additional changes for improved clarity:

In addition to the edits based on reviewers’ comments, we have also made the following modifications to improve our manuscript.

- The following figures have been regenerated to display corresponding data points, the specific sample sizes or error bars, Figures 2B, 3C, 4A,C, 5A,D-F, S5B-D, S7A, S8C, S13B,D, S16D.
- All error bars are defined and n numbers are given for relevant plots.
- We removed the ‘within-other-motif’ group from the comparison in Figure 3C since it is not well-defined. This change does not impact any statements or conclusions. The

updated Figure 3C aligns more closely with the comparisons in Figure S7A and other results on module analysis.

- We corrected Figure S8C and used the appropriate test. No statements or conclusions are changed.
- Figure S16, initially included for illustrative purposes, has been removed due to redundant information with Figure 2A.